# Scaling Diffusion Transformers Efficiently via $\mu$P

**Chenyu Zheng**[1,2,3,*] **Xinyu Zhang**[4] **, Rongzhen Wang**[1,2,3] **, Wei Huang**[5] **, Zhi Tian**[4] **,**
**Weilin Huang**[4] **, Jun Zhu**[6] **, Chongxuan Li**[1,2,3,†]

[1] Gaoling School of Artificial Intelligence, Renmin University of China
[2] Beijing Key Laboratory of Research on Large Models and Intelligent Governance
[3] Engineering Research Center of Next-Generation Intelligent Search and Recommendation, MOE
[4] ByteDance Seed  [5] RIKEN AIP  [6] Dept. of Comp. Sci. & Tech., Tsinghua University

## Abstract

Diffusion Transformers have emerged as the foundation for vision generative models, but their scalability is limited by the high cost of hyperparameter (HP) tuning at large scales. Recently, Maximal Update Parametrization ($\mu$P) was proposed for vanilla Transformers, which enables stable HP transfer from small to large language models, and dramatically reduces tuning costs. However, it remains unclear whether $\mu$P of vanilla Transformers extends to diffusion Transformers, which differ architecturally and objectively. In this work, we generalize standard $\mu$P to diffusion Transformers and validate its effectiveness through large-scale experiments. First, we rigorously prove that $\mu$P of mainstream diffusion Transformers, including DiT, U-ViT, PixArt-$\alpha$, and MMDiT, aligns with that of the vanilla Transformer, enabling the direct application of existing $\mu$P methodologies. Leveraging this result, we systematically demonstrate that DiT-$\mu$P enjoys robust HP transferability. Notably, DiT-XL-2-$\mu$P with transferred learning rate achieves $2.9\times$ faster convergence than the original DiT-XL-2. Finally, we validate the effectiveness of $\mu$P on text-to-image generation by scaling PixArt-$\alpha$ from 0.04B to 0.61B and MMDiT from 0.18B to 18B. In both cases, models under $\mu$P outperform their respective baselines while requiring small tuning cost—only $5.5\%$ of one training run for PixArt-$\alpha$ and $3\%$ of consumption by human experts for MMDiT-18B. *These results establish $\mu$P as a principled and efficient framework for scaling diffusion Transformers.*

## 1 Introduction

Owing to its generality and scalability, diffusion Transformers [52; 1] have become the backbone of modern vision generation models, with widespread applications in various tasks such as image generation [55; 3; 16; 35; 20] and video generation [6; 81; 62; 2; 77]. As datasets grow and task complexity increases, further scaling of diffusion Transformers has become inevitable and is attracting increasing attention [41; 40; 29; 17; 75]. However, as model sizes reach billions of parameters, hyperparameter (HP) tuning becomes prohibitively expensive, often hindering the model from achieving its full potential. This underscores the urgent need for a principled approach to efficiently identify the optimal HPs when scaling diffusion Transformers.

Maximal Update Parametrization ($\mu$P) [73; 70; 72] was recently proposed as a promising solution to the HP selection problem for large-scale vanilla Transformer [61]. It stabilizes optimal HPs across different model widths, enabling direct transfer of HPs searched from small models to large models (a.k.a., $\mu$Transfer) and significantly reducing tuning costs at scale. Due to its strong transferability, $\mu$P has been applied to the pretraining of large language models (LLMs) [73; 14; 28; 43; 42; 82].

---

*Work done during an internship at ByteDance Seed.

†Correspondence to Chongxuan Li <chongxuanli@ruc.edu.cn>.

39th Conference on Neural Information Processing Systems (NeurIPS 2025).

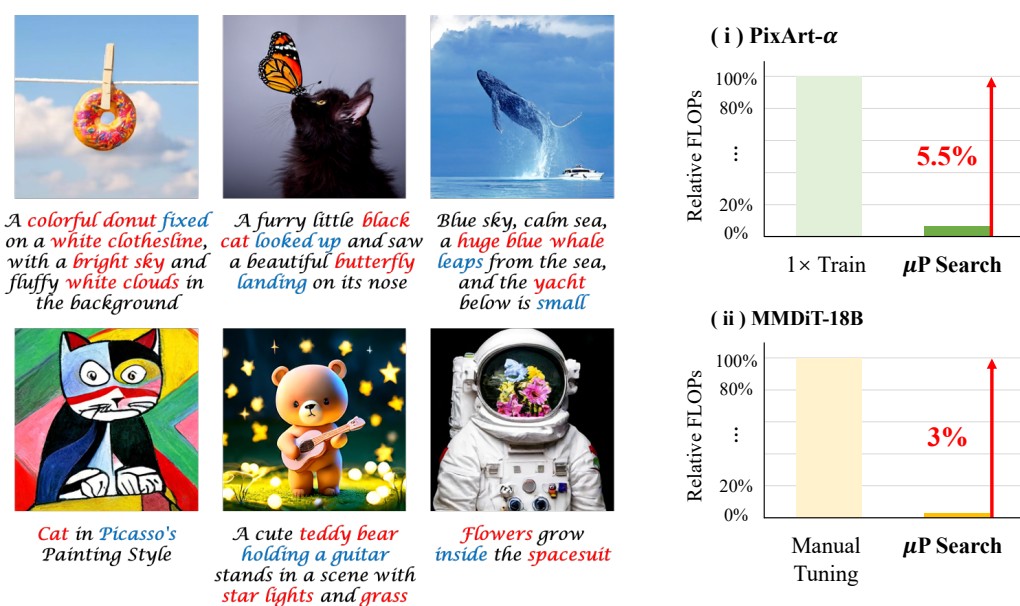

(a) Samples produced by the MMDiT-$\mu$P-18B.  (b) Efficiency of $\mu$P search.

Figure 1: **Visualization results and efficiency of HP search under $\mu$P.** (a) Samples generated by the MMDiT-$\mu$P-18B model exhibit strong fidelity and precision in aligning with the provided textual descriptions. (b) HP search for large diffusion Transformers is efficient under $\mu$P, requiring only $5.5\%$ FLOPs of a single training run for PixArt-$\alpha$ and just $3\%$ FLOPs of the human experts for MMDiT-18B.

Unfortunately, diffusion Transformers [52; 1; 16; 8] differ fundamentally from vanilla Transformers [61]. First, their architectures incorporate additional components to integrate text information and diffusion timesteps for final vision generation. Second, they operate under a distinct generative framework based on iterative denoising, in contrast to the autoregressive paradigms typically used in vanilla Transformers (e.g., LLMs) [22; 80]. As a result, existing $\mu$P theory and its associated HP transfer properties may not directly apply to diffusion Transformers. This paper systematically investigates this issue, as detailed below.

First, we extend the standard $\mu$P theory from vanilla Transformers to diffusion Transformers in Section 3.1. Using the Tensor Programs technique [72; 70; 67], we rigorously prove that the $\mu$P formulation of mainstream diffusion Transformers, including DiT [52], U-ViT [1], PixArt-$\alpha$ [8], and MMDiT from Stable Diffusion 3 [16], matches that of vanilla Transformers (Theorem 3.1). This compatibility enables us to directly apply existing practical methodologies developed for standard $\mu$P to diffusion Transformers, as described in Section 3.2.

Based on the rigorous $\mu$P result of diffusion Transformers, we then conduct a systematic study of DiT-$\mu$P on the image generation task using the ImageNet dataset [13], presented in Section 4. We first verify the stable HP transferability of DiT-$\mu$P across widths, batch sizes, and training steps. We then $\mu$Transfer the optimal learning rate searched from small models to DiT-XL-2-$\mu$P. Notably, DiT-XL-2-$\mu$P trained with the transferred learning rate achieves $2.9\times$ faster convergence than the original DiT-XL-2 [52], suggesting that $\mu$P offers an efficient principle for scaling diffusion Transformers.

Finally, we further validate the efficiency of $\mu$P on large-scale text-to-image generation tasks. In Section 5.1, we apply the $\mu$Transfer algorithm to PixArt-$\alpha$ [8], scaling the model from 0.04B to 0.61B. In Section 5.2, we apply it to MMDiT [16], scaling from 0.18B to 18B. In both cases, diffusion Transformers under $\mu$P consistently outperform their respective baselines with small HP tuning cost on proxy tasks. For PixArt-$\alpha$, tuning consumes only $5.5\%$ of the FLOPs required for a full pretraining run, while for MMDiT, tuning uses just $3\%$ of the FLOPs typically consumed by human experts. These real-world experiments further confirm the scalability and reliability of $\mu$P.

## 2 Preliminaries

We begin by establishing the necessary background for diffusion Transformers and $\mu$P. Detailed discussion of additional related work is placed in Appendix A.

Table 1: $\mu$**P for (diffusion) Transformers with Adam/AdamW optimizer.** We use purple text to highlight the differences between $\mu$P and standard parameterization (SP) in practice (e.g., Kaiming initialization [25]), and gray text to indicate the SP settings. Formal definitions of the weight type are provided in Appendix B.

|        | Input weights | Hidden weights | Output weights |
|--------|:-------------:|:--------------:|:--------------:|
| $a_W$  | 0             | 0              | 1 (0)          |
| $b_W$  | 0             | $1/2$          | 0 ($1/2$)      |
| $c_W$  | 0             | 1 (0)          | 0              |

## 2.1 Diffusion Transformers

Due to its superior scalability and compatibility, Transformers [52; 1; 16; 30] have replaced CNNs (e.g., U-Net [54; 53; 63]) as the backbone for advanced diffusion models [55; 3; 16; 35; 20]. U-ViT [1] firstly introduces Transformers with long skip connections between shallow and deep layers for various image generation tasks and achieves remarkable performance. After that, DiT [52] proposes Transformers with adaptive layer normalization (adaLN) blocks for class-to-image generation and demonstrates strong scalability with respect to network complexity. PixArt-$\alpha$ [8] extends DiT for text-conditional image generation by incorporating cross-attention for text features and an efficient shared adaLN-single block. MMDiT [16] further extends DiT by introducing two separate parameter sets for image and text modalities, along with a joint attention mechanism to facilitate multimodal interaction. As diffusion Transformers continue to scale, increasing attention is being paid to principled approaches for scaling [41; 40; 29; 75].

## 2.2 Maximal Update Parametrization

In this section, we provide a practical overview of $\mu$P, with a focus on the widely used AdamW optimizer [47]. The comprehensive review of the theoretical foundations of $\mu$P is in Appendix B.

$\mu$P identifies a unified parameterization that applies to common architectures expressive in NE⊗OR⊤ Program [72] (e.g., vanilla Transformer [61]), offering strong guidance for practical HP transfer across model widths, batch sizes, and training steps. Under $\mu$P, HPs can be tuned on a small proxy task (e.g., 0.04B parameters and 6B tokens in [73]) and directly transferred to a large-scale pretraining task (e.g., 6.7B parameters and 300B tokens in [73]), significantly reducing tuning costs at scale. As a result, $\mu$P has been widely adopted in the pretraining of LLMs [73; 14; 28; 43; 23; 42; 82].

Concretely, $\mu$P is implemented by analytically adjusting HPs with model width, typically involving the weight multiplier, initialization variance, and learning rate (a.k.a., $abc$-parameterization). Formally, let $n$ denote the network width, we set each weight as $\boldsymbol{W} = \phi_W n^{-a_W} \widetilde{\boldsymbol{W}}$, where the trainable component $\widetilde{\boldsymbol{W}}$ is initialized as $\widetilde{W}_{ij} \sim \mathcal{N}(0, \sigma_W^2 n^{-2b_W})$, and its learning rate is $\eta_W n^{-c_W}$. Henceforth, we call the width-independent parts ($\phi_W, \sigma_W, \eta_W$) as **base HPs**. As summarized in Table 1, $\mu$P identifies values of $a_W$, $b_W$, and $c_W$ that enable models of different widths to share the (approximately) **same optimal base HPs** $\phi_W^*, \sigma_W^*$, and $\eta_W^*$. $\mu$P adjusion for other HPs is placed in Appendix B.4.

The most related work to us is AuraFlow v0.1 [10], which applied $\mu$P empirically to an MMDiT-style model for learning rate transfer. Compared to AuraFlow v0.1, our work makes additional non-trivial contributions. First, we provide a rigorous theoretical proof for mainstream diffusion Transformers, thereby justifying the validity of $\mu$P for this family of models. Second, we systematically validate the HP transferability of diffusion Transformers under $\mu$P, across multiple widths, batch sizes, and training steps. Finally, for MMDiT in particular, we conduct a more extensive search over multiple HPs and scale the model up to 18B parameters, providing detailed intermediate results and comparisons.

## 3 Scaling Diffusion Transformers by $\mu$P

In this section, we extend the principles of $\mu$P to scale diffusion Transformers. In Section 3.1, we prove that despite fundamental differences from vanilla Transformer, mainstream diffusion Transformers—including U-ViT [1], DiT [52], PixArt-$\alpha$ [8], and MMDiT [16]—adhere to the standard $\mu$P formulation summarized in Table 1. Then, in Section 3.2, we introduce the practical methodology for applying $\mu$P to diffusion Transformers, as illustrated in Figure 2.

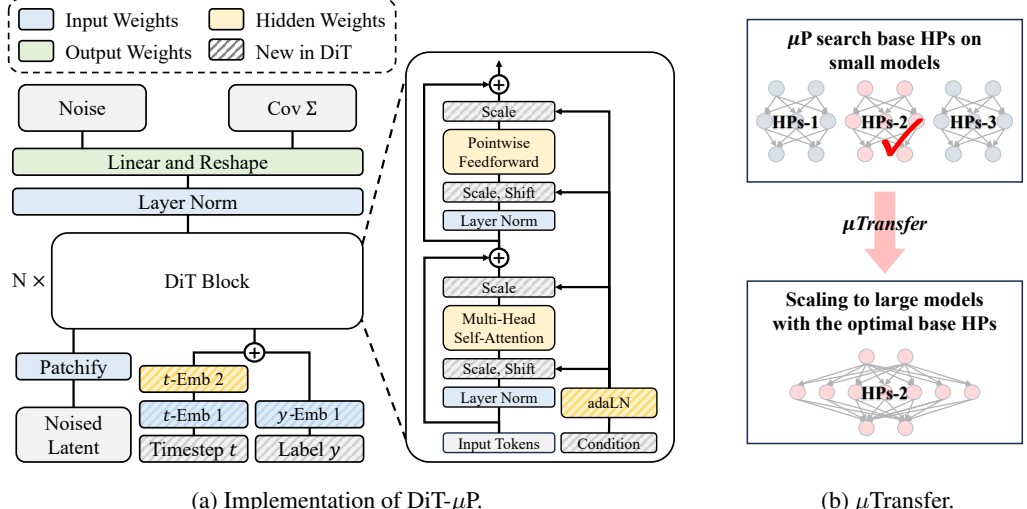

(a) Implementation of DiT-$\mu$P.           (b) $\mu$Transfer.

Figure 2: **A overview of applying $\mu$P to diffusion Transformers.** (a) We illustrate the implementation of $\mu$P for DiT as an example. The $abc$-parameterization of each weight is adjusted based on its type and visualized using different colors. Modules that differ from the vanilla Transformer are also highlighted. (b) We $\mu$Transfer the optimal base HPs searched from multiple trials on small models to pretrain the target large models.

## 3.1 $\mu$P of Diffusion Transformers

As mentioned in Section 2, the existing $\mu$P results [72; 73] in Table 1 apply only to architectures expressible as a $\mathrm{N E \otimes O R \top}$ program. Therefore, it is crucial to determine whether the diffusion Transformers can be represented in this framework. The following theorem establishes that several prominent diffusion Transformers [52; 1; 8; 16] are compatible with the existing $\mu$P results.

**Theorem 3.1** ($\mu$P of diffusion Transformers, proof in Appendix C). *The forward passes of mainstream diffusion Transformers (U-ViT [1], DiT [52], Pixart-$\alpha$ [8], and MMDiT [16]) can be represented within the $\mathrm{N E \otimes O R \top}$ Program. Therefore, their $\mu$P matches the standard $\mu$P presented in Table 1.*

The proof of Theorem 3.1 relies on rewriting the forward pass of diffusion Transformers using the three operators defined in the $\mathrm{N E \otimes O R \top}$ Program [72]. Given that the representability of standard Transformer [61] within the $\mathrm{N E \otimes O R \top}$ Program has been established [67], we focus on demonstrating that the novel modules specific to diffusion Transformers can also be expressed in this formalism. These modules primarily serve to integrate text and diffusion timestep information for vision generation; examples include the adaLN blocks of DiT [52] in Figure 2a.

Theorem 3.1 ensures that existing practical methodologies developed for standard $\mu$P apply directly to DiT, U-ViT, PixArt-$\alpha$, and MMDiT. Moreover, our analysis technique naturally extends to other variants of diffusion Transformers [49; 7; 19; 59; 48]. To the best of our knowledge, mainstream diffusion Transformer variants in use can be expressed within the $\mathrm{N E \otimes O R \top}$ Program.

## 3.2 Practical Methodology of $\mu$P in Diffusion Transformers

Given the rigorous $\mu$P result for diffusion Transformers established in Theorem 3.1, this section introduces how to apply $\mu$P to diffusion Transformers in practice. In addition, our code is available at *https://github.com/ML-GSAI/Scaling-Diffusion-Transformers-muP*.

### 3.2.1 Implementation of $\mu$P

The width of a multi-head attention layer is determined by the product of the head dimension and the number of attention heads. Thus, there are two degrees of freedom when scaling the width of diffusion Transformers. Theoretically, recent theoretical advances [5] reveal an important difference: when the head dimension tends to infinity, multi-head self-attention can collapse to single-head self-attention, losing the diversity of attention patterns. In contrast, scaling the number of heads avoids this degeneracy and preserves the expressivity of the multi-head mechanism. Empirically, well-known models in real-world large-scale practice [73; 22; 43; 28] favor increasing the number of

heads rather than the head dimension. Given these theoretical and empirical insights, we fix the head dimension and scale the number of heads in this work.

Given scaling the number of heads, we implement the $\mu$P of diffusion Transformers following the standard procedure in [73].[3] Specifically, we replace the vanilla width $n$ in the $abc$-parameterization (see Section 2) with the width ratio $n/n_{\text{base}}$ when standard parameterization (SP) and $\mu$P differ in Table 1, where $n_{\text{base}}$ is a fixed base width. This implementation remains consistent with Table 1 since $n_{\text{base}}$ is a constant. For example, for a hidden weight matrix $\boldsymbol{W}$ with width $n$ and base HPs $\phi_W, \sigma_W^2, \eta$ (maybe searched from proxy model), we set $\boldsymbol{W} = \phi_W \widetilde{\boldsymbol{W}}$ with initial $\widetilde{\boldsymbol{W}} \sim \mathcal{N}(0, \sigma_W^2/n)$, as in SP. In contrast, its learning rate is $\eta n_{\text{base}}/n$ rather than $\eta/n$, where $\mu$P differs from SP. Finally, we also follow the suggestion from [73] to initialize the output weights by zero (i.e., $\sigma_{\text{out}}^2 = 0$).

### 3.2.2 Base Hyperparameter Transferability of $\mu$P

Prior work has shown that $\mu$P provides strong guidance for base HP transferability in image classification and language modeling tasks [73; 28; 64; 4; 24; 60; 14]. Building on the rigorous $\mu$P form for diffusion Transformers in Theorem 3.1, we further validate its HP transferability in the context of visual generation. We summarize the methodology for verifying base HP transferability across widths, batch sizes, and training steps in Algorithm 1, 2, and 3 in Appendix D, respectively. We describe how to verify HP transfer across widths below; those for batch size and step are similar.

For simplicity, we describe how to evaluate the transferability of the base learning rate $\eta$ across widths; similar procedures apply to other HPs, such as the multiplier $\phi$. We define $\{n_i\}_{i=1}^P$ and $\{\eta_j\}_{j=1}^R$ as the sets of widths and base learning rates used in the evaluation. Given a fixed batch size $B$ and training iterations of $T$, we train $PR$ diffusion Transformers, each parameterized by $\mu$P with base width $n_{\text{base}}$, true width $n_i$, and base learning rate $\eta_j$. Finally, given evaluation metrics, if models at different widths (approximately) share the same optimal base learning rate, we conclude that learning rate transferability holds for diffusion Transformers. In Section 4, we verify that diffusion Transformers under the $\mu$P exhibit robust transferability of base HPs.

### 3.2.3 $\mu$Transfer from Proxy Task to Target Task

Once base HP transferability is validated for diffusion Transformers, we can directly apply the $\mu$Transfer algorithm [73] (see Algorithm 4 in Appendix D) to zero-shot transfer base HPs from a proxy task to a target task. Specifically, both the proxy model with width $n_{\text{proxy}}$ and the target model with width $n_{\text{target}}$ are parameterized by $\mu$P using the same base width $n_{\text{base}}$. We first search for an optimal combination of base HPs using various proxy models trained with a small batch size $B_{\text{proxy}}$ and a limited training steps $T_{\text{proxy}}$. These optimal base HPs are then used to train the large target model with a larger batch size $B_{\text{target}}$ and longer training iteration $T_{\text{target}}$. Experimental results in Section 5 show the superior performance of the $\mu$Transfer in real-world vision generation.

## 4 Systematic Investigation for DiT-$\mu$P on ImageNet

In this section, we first empirically verify the base HP transferability of DiT [52] under $\mu$P. We then $\mu$Transfer the optimal learning rate to train DiT-XL-2-$\mu$P, which achieves $2.9\times$ faster convergence.

### 4.1 Basic Experimental Settings

To ensure a fair comparison between DiT-$\mu$P and the original DiT [52], we adopt the default configurations from [52] and describe the basic setup in detail below.

**Dataset.** We train DiT and DiT-$\mu$P on the ImageNet training set [13], which contains 1.28M images across 1,000 classes. All images are resized to a resolution of $256 \times 256$ during training, following standard practice in generative modeling benchmarks [52; 1; 49].

**Architecture of DiT-$\mu$P.** The architecture of DiT-$\mu$P models is identical to that of DiT-XL-2, except for the width. We fix the attention head dimension at 72 (as in DiT-XL-2) and vary the number of heads. The base width $n_{\text{base}}$ in the $\mu$P setup corresponds to 288, which uses 4 attention heads.

---

[3]To scaling head dimension, Yang et al. [73] additionally change the calculation of attention logit from $\boldsymbol{q}^\top \boldsymbol{k}/\sqrt{d}$ to $\boldsymbol{q}^\top \boldsymbol{k}/d$, where query $\boldsymbol{q}$ and key $\boldsymbol{k}$ have dimension $d$. We do not use this modification [5].

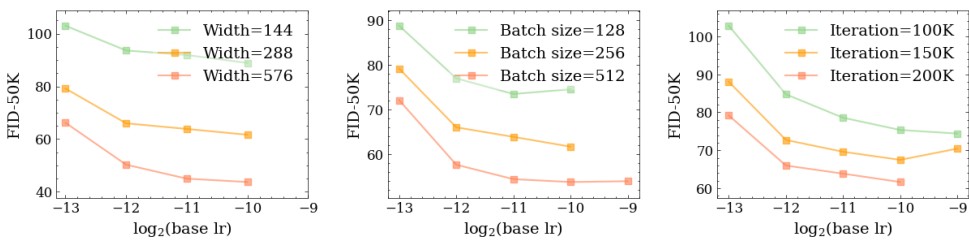

| (a) HP transfer across widths. | (b) HP transfer across batch sizes. | (c) HP transfer across iterations. |

Figure 3: **DiT-$\mu$P enjoys base HP transferability**. Unless otherwise specified, we use a model width of 288, a batch size of 256, and a training iteration of 200K. Missing data points indicate training instability, where the loss explodes. Under $\mu$P, the base learning rate can be transferred across model widths, batch sizes, and steps.

**Training.** We train DiT and DiT-$\mu$P using the AdamW [47]. Following the original DiT setup [52], we do not apply any learning rate schedule or weight decay, and constant learning rates are used in all experiments. The original DiT-XL-2 is trained with a learning rate $10^{-4}$ and a batch size of 256.

**Evaluation metrics.** To comprehensively evaluate generation performance, we report FID [26], sFID [50], Inception Score [56], precision, and recall [34] on 50K generated samples without classifier-free guidance (cfg), as in Table 4 of [52]. In the main text, we present the FID results, while the remaining metrics are provided in Appendix E.2.

### 4.2 Base Hyperparameters Transferability of DiT-$\mu$P

In this section, we evaluate the HP transferability of DiT-$\mu$P across different widths, batch sizes, and training steps. We focus primarily on the base learning rate, as it has the most significant impact on performance [73; 28; 14; 43]. *Similar results for other HP (weight multiplier) are provided in Appendix E.2*. We sweep the base learning rate over the set $\{2^{-13}, 2^{-12}, 2^{-11}, 2^{-10}, 2^{-9}\}$ across various widths, batch sizes, and training steps. FID-50K results are shown in Figure 3, and comprehensive results for other metrics are presented in Tables 8, 9, and 10 in Appendix E.2.

As presented in Figure 3, the optimal base learning rate $2^{-10}$ generally transfers across scaling dimensions when some minimum width (e.g. 144), batch size (e.g., 256), and training steps (e.g., 150K) are satisfied, which verifies the base HP transferability of DiT-$\mu$P. Interestingly, we observe that neural networks with $\mu$P tend to favor a large learning rate close to the maximum stable value (e.g., see Figure 3a). This aligns with empirical findings and theoretical insights reported for standard neural networks trained for multiple epochs [65; 44; 37; 38; 11], which suggest that larger learning rates introduce beneficial gradient noise to help guide optimization towards flatter minima that generalize better. Our empirical results suggest that the optimization landscape of $\mu$P shares certain similarities with that of SP, offering a direction for future theoretical investigation.

### 4.3 Scaling Performance of DiT-$\mu$P

We $\mu$Transfer the optimal base learning rate of $2^{-10}$ to train DiT-XL-2-$\mu$P with a width of 1152. The batch size is set to 256, following [52]. We evaluate DiT-$\mu$P every 400K steps without using cfg. Training continues until DiT-$\mu$P surpasses the best performance reported by the original DiT at final 7M steps [52]. To enable a detailed comparison throughout the training, we also reproduce the original DiT training using its official codebase. Complete evaluation results throughout training are provided in Table 14 in Appendix E.2.

As shown in Figure 4, DiT-XL-2-$\mu$P with the transferred base learning rate performs effectively, achieving 2.9× faster convergence (2.4M steps) compared to the original DiT (7M steps). These results suggest that $\mu$P offers a simple and promising approach to improve the pretraining of large-scale diffusion Transformers. In the following sections, we further validate this claim through text-to-image generation tasks.

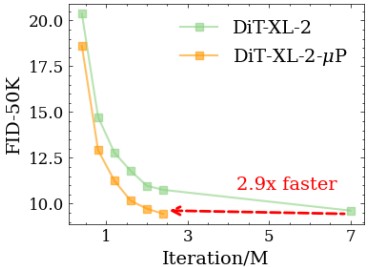

Figure 4: **$\mu$P accerlates the training of diffusion Transformers.** Considering FID-50K, DiT-XL-2-$\mu$P with transferred learning rate achieves 2.9× faster convergence than the original DiT-XL-2 and a slightly better result.

## 5 Large-Scale Text-to-Image Generation

In this section, we verify the efficiency of $\mu$Transfer algorithm on real-world text-to-image generation tasks. Diffusion Transformers under $\mu$P outperform the baselines while requiring small tuning cost.

### 5.1 Scaling PixArt-$\alpha$-$\mu$P on SA-1B

In this section, we perform $\mu$Transfer experiments on the PixArt-$\alpha$ [8], scaling from 0.04B to 0.61B parameters. Using the same pretraining setup, PixArt-$\alpha$-$\mu$P with the transferred learning rate outperforms the original PixArt-$\alpha$, while incurring only $5.5\%$ FLOPs of one full pretraining run.

#### 5.1.1 Experimental Settings

To ensure the fairest possible comparison between PixArt-$\alpha$-$\mu$P and the original PixArt-$\alpha$ [8], we mainly adopt the original setup [8] and summarize the key components below.

**Dataset.** We use the SAM/SA-1B dataset [33], which contains 11M high-quality images curated for segmentation tasks with diverse object-rich scenes. For text captions, we use the SAM-LLaVA annotations released in [8]. All images are resized to a resolution of $256 \times 256$ during training.

**Architecture of PixArt-$\alpha$-$\mu$P models.** The target PixArt-$\alpha$-$\mu$P model adopts the same architecture as PixArt-$\alpha$ (0.61B parameters). The proxy model also follows the same architecture, differing only in width. To construct the proxy PixArt-$\alpha$-$\mu$P model, we fix the attention head dimension at 72 (as in PixArt-$\alpha$) and reduce the number of heads from 16 to 4 (0.04B parameters). In the $\mu$P framework, the base width $n_{\text{base}}$ (see Section 3.2.1) is set to the proxy width of 288.

**Training.** PixArt-$\alpha$ is implemented using the official codebase and original configuration. We train the original PixArt-$\alpha$ and the target PixArt-$\alpha$-$\mu$P model for 30 epochs with a batch size of $176 \times 32$ (approximately 59K steps).[4] The small proxy PixArt-$\alpha$-$\mu$P models are trained for 5 epochs with a batch size of $176 \times 8$ (approximately 39K steps). *Notably, the model width, batch size, and training steps are all smaller than those used in the target pretraining setting.*

**Hyperparameter search.** In this section, we focus solely on the base learning rate. We search over the candidate set $\{2^{-13}, 2^{-12}, 2^{-11}, 2^{-10}, 2^{-9}\}$ as in Section 4, resulting in five proxy training trials.

**Ratio of tuning cost to pretraining cost.** We consider the FLOPs as the metric for the computational cost, then the ratio of tuning cost to pretraining cost can be estimated as

$$\texttt{ratio} = \frac{RS_{\text{proxy}}E_{\text{proxy}}}{S_{\text{target}}E_{\text{target}}} = \frac{RS_{\text{proxy}}B_{\text{proxy}}T_{\text{proxy}}}{S_{\text{target}}B_{\text{target}}T_{\text{target}}}, \tag{1}$$

where $R$ is the number of HP search trials, $S$ is the number of parameters, $E$ is the training epochs, $B$ is the batch size and $T$ is the training iteration. The $\texttt{ratio} \approx 5.5\%$ here, as detailed in Appendix E.3.

**Evaluation metrics.** We evaluate text-to-image generation performance following standard practice [8; 16; 66; 1], including FID, CLIP Score, and GenEval [21]. Both FID and CLIP Score are computed on the aesthetic MJHQ-30K [39] and real MS-COCO-30K [45] datasets. MJHQ-30K contains 30K images generated by Midjourney, while MS-COCO-30K is a randomly sampled subset of the MS-COCO [45] dataset. GenEval evaluates text-image alignment using 533 test prompts. Following the official implementation [8], we use a cfg of 4.5 to generate samples.

#### 5.1.2 Experimental Results of PixArt-$\alpha$-$\mu$P

We begin by conducting a base learning rate search using the PixArt-$\alpha$-$\mu$P proxy models. The evaluation results for different base learning rates are summarized in Table 2. Details of GenEval results can be found in Table 15 in Appendix E.3. Since overfitting is not observed in this setting, we include training loss as an additional evaluation metric. Overall, the base learning rate of $2^{-10}$ yields the best performance. Interestingly, this optimal learning rate matches that of DiT-$\mu$P (see Figure 3a). We hypothesize that this consistency arises from the architectural similarity between the two models and the fact that both the ImageNet and SAM datasets consist of real-world images. This observation suggests that optimal base HPs may exhibit some degree of transferability across different datasets and architectures.

---

[4]We confirmed with the authors that this setup is reasonable; longer training may result in overfitting.

Table 2: **Results of base learning rate search on PixArt-$\alpha$-$\mu$P proxy tasks.** 0.04B proxy models with different base learning rates are trained for 5 epochs on the SAM dataset. Overall, the base learning rate $2^{-10}$ is optimal.

| $\log_2(\text{lr})$ | Training loss ↓ | GenEval ↑ | FID-30K (MS-COCO) ↓ | FID-30K (MJHQ) ↓ |
|---|---|---|---|---|
| -9 | NaN | NaN | NaN | NaN |
| -10 | **0.1493** | **0.083** | **47.46** | 47.71 |
| -11 | 0.1494 | 0.078 | 49.24 | **46.31** |
| -12 | 0.1504 | 0.030 | 66.77 | 63.36 |
| -13 | 0.1536 | 0.051 | 60.28 | 60.93 |

Table 3: **Comprehensive comparison between PixArt-$\alpha$-$\mu$P and PixArt-$\alpha$.** Both models are trained on the SAM dataset for 30 epochs. PixArt-$\alpha$-$\mu$P (0.61B), using a base learning rate transferred from the optimal 0.04B proxy model, consistently outperforms the original baseline throughout the training process.

| Epoch | Method | GenEval ↑ | MJHQ | | MS-COCO | |
|---|---|---|---|---|---|---|
| | | | FID-30K ↓ | CLIP Score ↑ | FID-30K ↓ | CLIP Score ↑ |
| 10 | PixArt-$\alpha$ [8] | 0.19 | 38.36 | 25.78 | 34.58 | 28.12 |
| | PixArt-$\alpha$-$\mu$P (**Ours**) | **0.20** | **33.35** | **26.25** | **29.68** | **28.87** |
| 20 | PixArt-$\alpha$ [8] | 0.20 | 35.68 | 26.54 | 30.13 | 28.81 |
| | PixArt-$\alpha$-$\mu$P (**Ours**) | **0.23** | **33.42** | **26.83** | **29.05** | **29.53** |
| 30 | PixArt-$\alpha$ [8] | 0.15 | 42.71 | 26.25 | 37.61 | 28.91 |
| | PixArt-$\alpha$-$\mu$P (**Ours**) | **0.26** | **29.96** | **27.13** | **25.84** | **29.58** |

We then apply $\mu$Transfer by transferring the searched optimal base learning rate of $2^{-10}$ to train the target PixArt-$\alpha$. A comparison between PixArt-$\alpha$ and PixArt-$\alpha$-$\mu$P throughout training is provided in Table 3 (with the complete results shown in Table 16 in Appendix E.3). The results demonstrate that PixArt-$\alpha$-$\mu$P consistently outperforms PixArt-$\alpha$ across all evaluation metrics during training, supporting $\mu$P as an efficient and robust approach for scaling diffusion Transformers. Furthermore, we observe that the benchmark performance of PixArt-$\alpha$ degrades after 20 epochs, primarily due to overfitting. In contrast, PixArt-$\alpha$-$\mu$P continues to improve, suggesting that $\mu$P enhances the model's generalization ability, which offers an interesting direction for future theoretical investigation.

Specifically, we emphasize that our current experimental results validate that $\mu$P works well without cfg (DiT) and with cfg (PixArt-$\alpha$). Because cfg is an important factor during inference and affects the optimal hyperparameters, we strongly recommend practitioners align the evaluation of proxy models and that of the target model. In the following section, we further extend the current method to large-scale applications.

## 5.2 Scaling MMDiT to 18B

In this section, we validate the efficiency of $\mu$P in the large-scale setup. We scale up the MMDiT [16] architecture from 0.18B to 18B. Under the same pretraining setup, MMDiT-$\mu$P-18B with the transferred base HPs outperforms the MMDiT-18B tuned by human algorithmic experts.

### 5.2.1 Experimental Settings

**Dataset.** We train models on an internally constructed dataset comprising 820M high-quality image-text pairs. All images are resized to a resolution of $256 \times 256$ during training.

**Baseline MMDiT-18B.** The width and depth of MMDiT-18B are 5,120 and 16, respectively. The training objective combines a flow matching loss [46; 49] and a representation alignment (REPA) loss [76]. The model is optimized by AdamW [47], with a batch size of 4,096 and 200K training iterations. The learning rate schedule is constant with a warm-up duration. The HPs were tuned by algorithmic experts, requiring roughly 5 times the cost of full pretraining, as detailed in Appendix E.4.

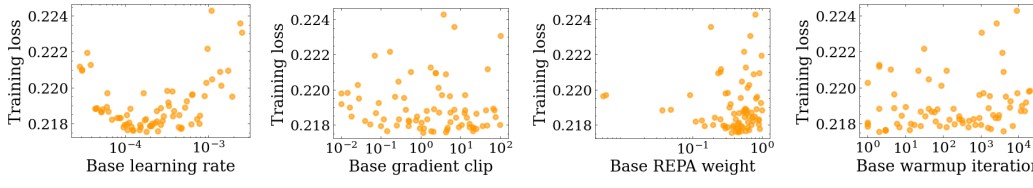

(a) Base learning rate.    (b) Base gradient clip.    (c) Base REPA loss weight.    (d) Base Warm-up steps.

Figure 5: **Results of base HP search on proxy MMDiT-$\mu$P tasks.** We train 0.18B MMDiT-$\mu$P proxy models with 80 different base HPs settings. The optimal base HPs are transferred to the training of 18B target model.

**Architecture of MMDiT-$\mu$P models.** The target MMDiT-$\mu$P-18B model shares the same architecture as MMDiT-18B. The proxy model also follows this architecture, differing only in width by reducing the number of attention heads. It contains 0.18B parameters ($1\%$ **of the target model**) with a width of 512. In the $\mu$P setup, the base width $n_{\text{base}}$ is set to 1,920 (see Section 3.2.1 for definition of $n_{\text{base}}$).

**Training of MMDiT-$\mu$P models.** The training procedure for the target MMDiT-$\mu$P-18B is identical to that of the baseline MMDiT-18B models, except for the selected HPs. The proxy models are trained for 30K steps with a batch size of 4,096. We conduct 80 searches over four base HPs. Concretely, we uniformly sample base learning rate from $2.5 \times 10^{-5}$ to $2.5 \times 10^{-3}$, gradient clipping from 0.01 to 100, weight of REPA loss from 0.1 to 1, and warm-up iteration from 1 to 20K. To further verify that 30K iterations are enough for the proxy task, we conducted five proxy training runs (100K steps each) using the searched optimal HPs and different learning rates, as detailed in Appendix E.4.

As derived in Appendix E.4 based on Equation (1), the total tuning FLOPs under $\mu$P amounts to $14.5\%$ of one full pretraining cost, and thus only $3\%$ of the human-tuned cost. We think the batch size and iterations used during the base HP search could be further reduced to lower the tuning cost. However, due to limited resources, we are unable to explore additional setups in this work.

**Evaluation metrics.** We use training loss to select base HPs on proxy tasks, as the dataset is passed through only once, and overfitting does not occur. We follow the standard practice in the µP literature [73; 28; 14], where the base hyperparameter is typically selected by identifying the value that aligns with the lowest envelope of the training loss plot. To assess the final text-to-image generation, in addition to the standard GenEval benchmark [21], we also created an internal benchmark comprising 150 prompts to comprehensively evaluate text-image alignment. Each prompt includes an average of five binary alignment checks (yes or no), covering a wide range of concepts such as nouns, verbs, adjectives (e.g., size, color, style), and relational terms (e.g., spatial relationships, size comparisons). Ten human experts conducted the evaluation, with each prompt generating three images, resulting in a total of 22,500 alignment tests. The final score is computed as the average correctness across all test points. The details can be found in Appendix E.4.

### 5.2.2 Analyzing the Results of the Random Search

The visualization of the results of the base HP search is shown in Figure 5. First, the base learning rate has the most significant impact on training loss. In our case, we observed that the envelope near $2.5 \times 10^{-4}$ was stable and close to optimal, so we chose $2.5 \times 10^{-4}$ as the optimal value. Interestingly, unlike the DiT and PixArt setups, the optimal learning rate here is not close to the maximum stable value. This highlights a key difference between multi-epoch training in traditional deep learning and the single-epoch training in large model pretraining [65]. Intuitively, since the gradient signal for any individual sample is not revisited, the training must be more conservative to maintain stability. Second, the optimal gradient clipping value is 1, which deviates from the common practice of using a small value (e.g., 0.1) to stabilize pretraining. Intuitively, $\mu$P favors a larger clipping value, as aggressive clipping can undermine the maximal update property central to $\mu$P. Third, the optimal weight for the REPA loss is determined to be around 0.5, consistent with the experience from existing works [76]. Finally, the warm-up iteration has a negligible impact on the training loss, so we adopt the default value of 1K in the pretraining of the target model.

### 5.2.3 $\mu$Transfer Results of MMDiT-$\mu$P

The comparisons of training losses, GenEval results, and human evaluations are shown in Figure 6, Table 4, and Table 5, respectively. In addition, the complete comparison of training losses and detailed

Table 4: **GenEval results of pretrained MMDiT-18B and MMDiT-$\mu$P-18B models.** MMDiT-$\mu$P-18B achieves better benchmark results with only $3\%$ of the manual tuning cost.

| Method | Overall ↑ | Single | Two | Counting | Colors | Position | Color Attribution |
|--------|-----------|--------|-----|----------|--------|----------|-------------------|
| MMDiT-18B | 0.8154 | 99.38 | 93.69 | **81.88** | 88.03 | 57.5 | **68.75** |
| MMDiT-$\mu$P-18B | **0.8218** | 99.38 | **94.44** | 79.69 | **88.83** | **62.25** | 68.5 |

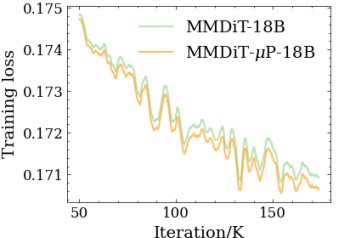

Figure 6: **MMDiT-$\mu$P-18B achieves consistently lower training loss** than baseline after 15K steps.

Table 5: **Results of human evaluation for text-image alignment.** The alignment accuracy (acc.) is computed as the average over 22,500 human alignment tests. MMDiT-$\mu$P-18B achieves superior results with only $3\%$ of the manual tuning cost.

| Method | Alignment acc. ↑ |
|--------|------------------|
| MMDiT-18B | 0.703 |
| MMDiT-$\mu$P-18B (**Ours**) | **0.715** |

visualizations are provided in Figure 8 and Figure 10 in Appendix E.4. As a result, MMDiT-$\mu$P-18B outperforms the baseline MMDiT-18B in all cases, achieving this with only $3\%$ FLOPs of the standard manual tuning cost. These results demonstrate that $\mu$P is a reliable principle for scaling diffusion Transformers. *As models grow larger and standard HP tuning becomes prohibitively expensive, $\mu$P offers a scientifically grounded framework to unlock the full potential of large models.*

## 6   Discussion

There are several promising research directions building on this work. First, the principles of $\mu$P could be extended to diffusion models with more advanced and efficient architectures, such as linear transformers [66] and mixture-of-expert models [17]. Second, $\mu$P can be applied to more sophisticated optimization algorithms, including the Muon optimizer [31] and the warmup-stable-decay learning schedule [28]. Third, while our experiments suggest that a proxy model width of 256–512 and a proxy dataset size of 1/10–1/6 of the full pretraining data are sufficient for stable HP transfer in diffusion Transformers, determining the optimal proxy task size that balances tuning cost and target model performance remains an important avenue for future work. Finally, developing a learning-theoretic framework to explain the optimization dynamics [58; 51], generalization behavior [9], and downstream performance [79; 78] of diffusion Transformers under $\mu$P would be both meaningful and impactful. In summary, this provides the vision community with an initial principled approach for scaling diffusion Transformers efficiently.

**Broader Impacts and Limitations.** Our work has the potential to accelerate progress in generative modeling applications using diffusion Transformers, including text-to-image and video generation. However, improvements in scaling diffusion Transformers could also facilitate the creation of deepfakes for disinformation. Regarding the limitations of this work, although we demonstrate the efficiency of $\mu$P in large-scale applications, we do not identify the optimal proxy task size that balances HP tuning cost and target model performance, due to limited computational resources.

## 7   Conclusion

In this paper, we extend $\mu$P from standard Transformers to diffusion Transformers. By proving that mainstream diffusion Transformers share the same $\mu$P form as vanilla Transformers, we enable direct application of existing $\mu$P practice and verify the reliable base HP transfer from small to large diffusion Transformers. This leads to practical performance gains on DiT-XL-2, PixArt-$\alpha$, and MMDiT-18B, while requiring a small fraction of the usual tuning effort (e.g., $3\%$ for MMDiT-18B). Our results establish $\mu$P as a principled and efficient scaling strategy for diffusion Transformers.

## Acknowledgments

This work was supported by the National Natural Science Foundation of China (No. 92470118); the Beijing Natural Science Foundation (No. L247030); the Beijing Major Science and Technology Project under Contract No. Z251100008425002; the Beijing Nova Program (No. 20230484416); the ByteDance Seed Research Fund; the Public Computing Cloud of Renmin University of China; and the fund for building world-class universities (disciplines) of Renmin University of China. Chenyu Zheng was also supported by the Outstanding Innovative Talents Cultivation Funded Programs 2024 of Renmin University of China. Finally, the authors thank Enze Xie for his helpful discussion on the experimental setup of PixArt-$\alpha$.

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

# Contents of Appendix

# Appendix A  Additional Related Work

## A.1  Scaling Diffusion Transformers

HourglassDiT [12] introduces a hierarchical architecture with downsampling and upsampling in DiT, reducing computational complexity for high-resolution image generation. SD3 [16] scales MMDiT to 8B parameters by increasing the backbone depth. Large-DiT [18] incorporates LLaMA's text embeddings and scales the DiT backbone, demonstrating that scaling to 7B parameters improves convergence speed. DiT-MoE [17] further scales diffusion Transformers to 16.5B parameters using a mixture-of-experts architecture. Li et al. [40] empirically investigate the scaling behavior of various diffusion Transformers, finding that U-ViT [1] scales more efficiently than cross-attention-based DiT variants. Yin et al. [75] systematically analyze scaling laws for video diffusion Transformers, enabling the prediction of optimal hyperparameters (HPs) for any model size and compute budget. We note that all these works adopt standard parameterization (SP), which prevents the transfer of optimal HPs across different model sizes, and thus suffer from heavy tuning costs at scale.

## A.2  Applications of $\mu$P in AIGC

Recently, $\mu$P has been successfully applied to the pretraining of large language models (LLMs) [73; 14; 28; 43; 23; 42; 82], reducing HP tuning costs and stabilizing training. For diffusion Transformers, in addition to AuraFlow v0.1 [10], some works have also employed $\mu$P. For example, Gulrajani and Hashimoto [23] searches for the base learning rate under $\mu$P and transfers it to a 1B-parameter diffusion language model, while Sargent et al. [57] uses $\mu$P to tune the HPs of a Transformer-based diffusion autoencoder, achieving state-of-the-art image tokenization across multiple compression rates. However, these studies differ fundamentally from ours. First, while they assume the HP transferability property of original $\mu$P (even though it may not hold in their setups), our work provides rigorous theoretical guarantees for the $\mu$P of diffusion Transformers and systematically verifies its HP transferability. Second, we validate the scalability of $\mu$P at up to 18B parameters, significantly larger than the models used in prior work.

# Appendix B  Theoretical Background of $\mu$P

In this section, we review the theoretical foundations of $\mu$P. For simplicity, we mainly focus on the Adam/AdamW optimizer widely used in practice. Besides, the $\epsilon$ in the optimizer is assumed to be zero, which is very small in practice (e.g., $10^{-8}$).

## B.1  $abc$-Parameterization Setup

For completeness, we restate the $abc$-parameterization from Section 2, which underpins typical $\mu$P theory. The $abc$-parameterization specifies the multiplier, initial variance, and learning rate for each weight [70; 73]. Specifically, let $n$ denote the network width, we parameterize each weight as $\boldsymbol{W} = \phi_W n^{-a_W} \widetilde{\boldsymbol{W}}$, where the trainable component $\widetilde{\boldsymbol{W}}$ is initialized as $\widetilde{W}_{ij} \sim \mathcal{N}(0, \sigma_W^2 n^{-2b_W})$. The learning rate for $\widetilde{\boldsymbol{W}}$ is parameterized as $\eta_W n^{-c_W}$. The width-independent quantities $\phi_W$, $\sigma_W$, and $\eta_W$ are referred to as base HPs and can be transferred across different widths.

## B.2  NE⊗ORT Program

$\mu$P is originally proposed by Tensor Programs [67; 68; 71; 69; 70; 72; 74], which is a theoretical series mainly developed to express and analyze the infinite-width limits of neural networks both at initialization and during training. The latest and most general version among them is the NE⊗ORT Program [72], which is formed by inductively generating scalars and vectors, starting from initial matrices/vectors/scalars (typically the initial weights of the neural network) and applying defined operations, including vector averaging (`Avg`), matrix multiplication (`MatMul`), and nonlinear outer product transformations (`OuterNonlin`). We introduce these operators, where $n$ is the model width tending to be infinite.

Table 6: $\mu$P for (diffusion) Transformers with Adam/AdamW optimizer. We use purple text to highlight the differences between $\mu$P and standard parameterization (SP) in practice (e.g., Kaiming initialization [25]), and gray text to indicate the SP settings. We do not contain the scalar weights in the main text because they do not exist in DiT.

|       | Input weights | Hidden weights | Output weights | Scalar weights |
|-------|:---:|:---:|:---:|:---:|
| $a_W$ | 0 | 0 | 1 (0) | 0 |
| $b_W$ | 0 | $1/2$ | 0 ($1/2$) | 0 |
| $c_W$ | 0 | 1 (0) | 0 | 0 |

**Avg** We can choose a existing vector $\boldsymbol{x} \in \mathbb{R}^n$ and append to the program a scalar

$$\frac{1}{n}\sum_{\alpha=1}^{n} x_\alpha \in \mathbb{R}.$$

**MatMul** We can choose a matrix $\boldsymbol{W} \in \mathbb{R}^{n \times n}$ and vector $\boldsymbol{x} \in \mathbb{R}^n$ existing in the program, and append to the program a vector

$$\boldsymbol{W}\boldsymbol{x} \in \mathbb{R}^n \quad \text{or} \quad \boldsymbol{W}^\top \boldsymbol{x} \in \mathbb{R}^n.$$

**OuterNonlin** For any integer $k, l \geq 0$, we can aggregate $k$ existing vectors and $l$ existing scalars to $\boldsymbol{X} \in \mathbb{R}^{n \times k}$ and $\boldsymbol{c} \in \mathbb{R}^l$, respectively. With an integer $r \geq 0$ and a pseudo-Lipschitz function $\psi : \mathbb{R}^{k(r+1)+l} \to \mathbb{R}$ (e.g., SiLU, GeLU), we append to the program a vector

$$\boldsymbol{y} \in \mathbb{R}^n, \quad y_\alpha = \frac{1}{n^r}\sum_{\beta_1,\ldots,\beta_r=1}^{n} \psi(\boldsymbol{X}_{\alpha:}; \boldsymbol{X}_{\beta_1:}; \ldots; \boldsymbol{X}_{\beta_r:}; \boldsymbol{c}^\top),$$

where the $r + 1$ is called the *order* of $\psi$.

While the above operators do not directly instruct to transform scalars into scalars, previous work [72] has proved that this can be done by combining the above operators. This can be formed as follows.

**Lemma B.1** (Scalars-to-scalars transformation is representable by the NE$\otimes$OR$\top$ Program, Lemma 2.6.2 in [72]). *If $\psi : \mathbb{R}^l \to \mathbb{R}$ is a pseudo-Lipschitz function and $\boldsymbol{c}$ are the collected scalars in a NE$\otimes$OR$\top$ program, then $\psi(\boldsymbol{c}) \in \mathbb{R}$ can be introduced as a new scalar in the program.*

## B.3 $\mu$P of Any Representable Archtecture

The seminal work of Yang and Littwin [72] systematically studies the dynamics of common neural architectures trained with adaptive optimizers (e.g., Adam [32], AdamW [47]), where the forward pass of neural network can be expressed as a NE$\otimes$OR$\top$ program (e.g., MLPs, CNNs [36], RNNs [27], standard Transformers [61]). In this setting, Yang and Littwin [72] prove the existence of a unique, optimal scaling rule—Maximal Update Parametrization ($\mu$P)—under which features in each layer evolve independently of width and at maximal strength. Otherwise, feature evolution in some layers will either explode or vanish in the infinite-width limit, rendering large-scale pretraining ineffective.

Concretely, $\mu$P is implemented by analytically adjusting HPs with model width, typically involving the weight multiplier, initialization variance, and learning rate considered in $abc$-parameterization. The $\mu$P formulation of any architecture whose forward pass can be expressed as a NE$\otimes$OR$\top$ program follows the same scaling rules presented in Table 6 (equivalent to Table 1 in the main text). Specifically, it adjusts the $abc$-parameterization of each weight according to its type, where the weights are categorized into four types: input weights, hidden weights, output weights, and scalar weights. The definitions are as follows: input weights satisfy $d_{\text{in}} = \Theta(1)$ and $d_{\text{out}} = \Theta(n)$; hidden weights satisfy $d_{\text{in}} = \Theta(n)$ and $d_{\text{out}} = \Theta(n)$; output weights satisfy $d_{\text{in}} = \Theta(n)$ and $d_{\text{out}} = \Theta(1)$; and scalar weights satisfy $d_{\text{in}} = \Theta(1)$ and $d_{\text{out}} = \Theta(1)$, where $n, d_{\text{in}}, d_{\text{out}}$ denotes the model width, fan-in dimension and fan-out dimension, respectively. The $\mu$P result is formally summarized in the following lemma.

**Lemma B.2** ($\mu$P of representable architecture, Definition 2.9.12 in [72]). *If the forward pass of an architecture can be represented by a NE$\otimes$OR$\top$ program, its $\mu$P formulation follows the scaling rules in Table 6 (equivalent to Table 1 in the main text).*

## B.4 Extensions to Other HPs

The $\mu$P principle can be extended to other HPs, such as common optimization-related HPs (e.g., warm-up iteration). We refer the reader to [73; 72] for a comprehensive discussion. We mainly focus on the HPs that occur in the main text.

- Gradient clip: The clip value should be held constant with respect to width.
- Warm-up iteration: The warm-up iteration should be held constant with respect to width.
- Weights of different losses (e.g., REPA loss, noise schedule): The weights of different losses should be held constant with respect to width.
- Weight decay: For coupled weight decay (e.g., Adam [32]), the weight decay value can not be transferred. For decoupled weight decay (e.g., AdamW [47]), the weight decay value should be held constant with respect to width.

# Appendix C  Proof of Theorem 3.1

**Elementary notations.** We use lightface (e.g., a, A), lowercase boldface (e.g., $\boldsymbol{a}$), and uppercase boldface letters (e.g., $\boldsymbol{A}$) to denote scalars, vectors, and matrices, respectively. For a vector $\boldsymbol{a}$, we denote its $i$-th element as $a_i$. For a matrix $\boldsymbol{A}$, we use $\boldsymbol{A}_{k:}$, $\boldsymbol{A}_{:k}$ and $A_{ij}$ to denote its $k$-th row, $k$-th column and $(i,j)$-th element, respectively. We define $[n] = \{1, 2, \ldots, n\}$.

For the reader's convenience, we restate Theorem 3.1 as follows.

**Theorem C.1.** *The forward passes of mainstream diffusion Transformers (U-ViT [1], DiT [52], Pixart-$\alpha$ [8], and MMDiT [16]) can be represented within the $\mathrm{N}\mathrm{E}\otimes\mathrm{O}\mathrm{R}\top$ Program. Therefore, their $\mu$P matches the standard $\mu$P presented in Table 6 (equivalent to Table 1 in the main text).*

Following the standard practice in the literature [70; 72; 74; 24] to simplify the derivation, we adopt the following assumption in all of our proofs.

**Assumption C.1.** *We assume that constant dimensions (fixed during scaling the width of neural networks) are 1, which includes data dimension, label dimension, patch size, frequency embedding size of time embedding, dimension of attention head, and so on. We also assume the width of different layers is the same $n$ and do not consider bias parameters here.*

In principle, the proof technique naturally extends to any finite fixed dimension ($\geq 1$) and to hidden layers with different widths (non-uniform infinite-width setting) [67; 72; 24]. We refer the readers to Section 2.9 in Yang and Littwin [72], which provides a detailed discussion of both these generalizations and shows how the tensor program framework naturally accommodates them.

## C.1  Proof of DiT

*Proof.* By Lemma B.2, we can prove the theorem by proving the forward pass of DiT can be represented by the $\mathrm{N}\mathrm{E}\otimes\mathrm{O}\mathrm{R}\top$ Program.

The DiT architecture includes an embedder for the input latent $x \in \mathbb{R}$, an embedder for the diffusion timestep $t \in \mathbb{R}$, an embedder for the label $y \in \mathbb{R}$, Transformer blocks with adaLN, and a final layer with adaLN. In the following, we use the $\mathrm{N}\mathrm{E}\otimes\mathrm{O}\mathrm{R}\top$ Program to represent the forward computation of these modules in sequence.

**Embedder for the input latent** $x$. The embedder for the $x$ is a one-layer CNN, which is known to be representable by the $\mathrm{N}\mathrm{E}\otimes\mathrm{O}\mathrm{R}\top$ Program (e.g., see Program 7 in [67] for general CNNs with pooling). We write the program for clarity here. In our case, the one-layer CNN's parameters can be denoted by $\boldsymbol{w}^{\mathrm{CNN}} \in \mathbb{R}^n$, and the operation can be implemented by

$$x_\alpha^{\mathrm{embed}} := \psi(w_\alpha^{\mathrm{CNN}}; x), \quad (\texttt{OuterNonlin})$$

where $\psi(w_\alpha^{\mathrm{CNN}}; x) = w_\alpha^{\mathrm{CNN}} x$.

**Positional embedding.** Given a number of patches (1 in our simplified case), the positional embedding is initialized with the sine-cosine version and fixed during the training. We use the $\boldsymbol{x}^{\mathrm{pos}}$ to denote the positional embedding, and adding it to the $\boldsymbol{x}^{\mathrm{embed}}$ can be written as

$$x_\alpha^{\mathrm{embed}} := \psi([\boldsymbol{x}^{\mathrm{embed}}, \boldsymbol{x}^{\mathrm{pos}}]_{\alpha:}), \quad (\texttt{OuterNonlin})$$

where $\psi([\boldsymbol{x}^{\text{embed}}, \boldsymbol{x}^{\text{pos}}]_{\alpha:}) = x_\alpha^{\text{embed}} + x_\alpha^{\text{pos}}$. Here, we reuse the notation of $\boldsymbol{x}^{\text{embed}}$ of the final embedding for the input latent $x$ without confusion.

**Embedder for the diffusion timestep $t$.** The embedder for the diffusion timestep $t$ is a frequency embedding (one dimension here) followed by a two-layer MLP. First, for the frequency embedding, we can represent it as

$$t^{\text{freq}} := \psi(t). \quad \text{(Lemma B.1)}$$

Second, the MLP is known to be representable by the NE⊗OR⊤ Program (e.g., Program 1 in [67]). We write it here for completeness. We use $\boldsymbol{w}^{(1)} \in \mathbb{R}^n$ and $\boldsymbol{W}^{(2)} \in \mathbb{R}^{n \times n}$ to denote the weights in the first layer and the second layer of the MLP, respectively. We can derive

$$t_\alpha^{(1)} := \psi_1(w_\alpha^{(1)}; t^{\text{freq}}), \quad\quad\quad\quad \text{(OuterNonlin)}$$
$$h_\alpha^{(1)} := \psi_2(t_\alpha^{(1)}), \quad\quad\quad\quad\quad \text{(OuterNonlin)}$$
$$\boldsymbol{t}^{\text{embed}} := \boldsymbol{W}^{(2)} \boldsymbol{h}^{(1)}, \quad\quad\quad\quad \text{(MatMul)}$$

where $\psi_1(w_\alpha^{(1)}; t^{\text{freq}}) = w_\alpha^{(1)} t^{\text{freq}}$ and $\psi_2(t_\alpha^{(1)}) = \text{SiLU}(t_\alpha^{(1)})$.

**Embedder for the label $y$.** Denoting the weights of embedding as $\boldsymbol{w}^{\text{embed}} \in \mathbb{R}^n$, then its computation can be written directly as

$$y_\alpha^{\text{embed}} := \psi(w_\alpha^{\text{embed}}; y), \quad \text{(OuterNonlin)}$$

where $\psi(w_\alpha^{\text{embed}}; y) = w_\alpha^{\text{embed}} y$.

**Merging the embeddings of $t$ and $y$.** In DiT, we sum the embeddings of $t$ and $y$ as the condition $\boldsymbol{c}$ for the following calculation, which can be written as

$$c_\alpha := \psi([\boldsymbol{t}^{\text{embed}}, \boldsymbol{y}^{\text{embed}}]_{\alpha:}), \quad \text{(OuterNonlin)}$$

where $\psi([\boldsymbol{t}^{\text{embed}}, \boldsymbol{y}^{\text{embed}}]_{\alpha:}) = t_\alpha^{\text{embed}} + y_\alpha^{\text{embed}}$.

**Transformer block with adaLN.** First, the adaLN block maps the condition $\boldsymbol{c}$ to the gate, shift, and scale values $(\boldsymbol{\theta}^{\text{attn}}, \boldsymbol{\beta}^{\text{attn}}, \boldsymbol{\gamma}^{\text{attn}}, \boldsymbol{\theta}^{\text{mlp}}, \boldsymbol{\beta}^{\text{mlp}}, \boldsymbol{\gamma}^{\text{mlp}})$ for attention layer and MLP layer in the Transformer block. We denote the parameters of the adaLN block by $(\boldsymbol{W}^{\theta-\text{attn}}, \boldsymbol{W}^{\beta-\text{attn}}, \boldsymbol{W}^{\gamma-\text{attn}}, \boldsymbol{W}^{\theta-\text{mlp}}, \boldsymbol{W}^{\beta-\text{mlp}}, \boldsymbol{W}^{\gamma-\text{mlp}})$, each of them is in $\mathbb{R}^{n \times n}$. Then the forward pass of adaLN block can be written as

$$\widetilde{c}_\alpha := \psi(c_\alpha), \quad\quad\quad\quad\quad\quad \text{(OuterNonlin)}$$
$$\boldsymbol{\theta}^{\text{attn}} := \boldsymbol{W}^{\theta-\text{attn}} \widetilde{\boldsymbol{c}}, \quad\quad\quad\quad \text{(MatMul)}$$
$$\boldsymbol{\beta}^{\text{attn}} := \boldsymbol{W}^{\beta-\text{attn}} \widetilde{\boldsymbol{c}}, \quad\quad\quad\quad \text{(MatMul)}$$
$$\boldsymbol{\gamma}^{\text{attn}} := \boldsymbol{W}^{\gamma-\text{attn}} \widetilde{\boldsymbol{c}}, \quad\quad\quad\quad \text{(MatMul)}$$
$$\boldsymbol{\theta}^{\text{mlp}} := \boldsymbol{W}^{\theta-\text{mlp}} \widetilde{\boldsymbol{c}}, \quad\quad\quad\quad \text{(MatMul)}$$
$$\boldsymbol{\beta}^{\text{mlp}} := \boldsymbol{W}^{\beta-\text{mlp}} \widetilde{\boldsymbol{c}}, \quad\quad\quad\quad \text{(MatMul)}$$
$$\boldsymbol{\gamma}^{\text{mlp}} := \boldsymbol{W}^{\gamma-\text{mlp}} \widetilde{\boldsymbol{c}}, \quad\quad\quad\quad \text{(MatMul)}$$

where $\psi$ is the SiLU activation.

Now, we denote the input feature of the Transformer block as $\boldsymbol{x} \in \mathbb{R}^n$. A layer norm without learnable parameters first normalizes it, which is also known to be representable by the NE⊗OR⊤ Program (e.g., Program 9 in [67]). We rewrite it in our simplified case.

$$\mathbb{E}[\boldsymbol{x}] := \frac{1}{n} \sum_{\alpha=1}^n x_\alpha, \quad\quad\quad\quad \text{(Avg)}$$
$$\bar{x}_\alpha^2 := \psi_1(x_\alpha; \mathbb{E}[\boldsymbol{x}]), \quad\quad\quad\quad \text{(OuterNonlin)}$$
$$\mathbb{V}[\boldsymbol{x}] := \frac{1}{n} \sum_{\alpha=1}^n \bar{x}_\alpha^2, \quad\quad\quad\quad \text{(Avg)}$$
$$x_\alpha^{\text{norm}} := \psi_2(x_\alpha; [\mathbb{E}[\boldsymbol{x}], \mathbb{V}[\boldsymbol{x}]]), \quad\quad \text{(OuterNonlin)}$$

where $\psi_1(x_\alpha; \mathbb{E}[\boldsymbol{x}]) = (x_\alpha - \mathbb{E}[\boldsymbol{x}])^2$, $\psi_2(x_\alpha; [\mathbb{E}[\boldsymbol{x}], \mathbb{V}[\boldsymbol{x}]]) = (x_\alpha - \mathbb{E}[\boldsymbol{x}])/\sqrt{\mathbb{V}[\boldsymbol{x}] + \epsilon}$.

After that, the normalized feature interacts with the condition information via the output of adaLN ($\boldsymbol{\beta}^{\mathrm{attn}}, \boldsymbol{\gamma}^{\mathrm{attn}}$), which can be represented as follows.

$$\widetilde{x}_\alpha^{\mathrm{norm}} := \psi([\boldsymbol{x}^{\mathrm{norm}}, \boldsymbol{\beta}^{\mathrm{attn}}, \boldsymbol{\gamma}^{\mathrm{attn}}]_{\alpha:}), \quad (\texttt{OuterNonlin})$$

where $\psi([\boldsymbol{x}^{\mathrm{norm}}, \boldsymbol{\beta}^{\mathrm{attn}}, \boldsymbol{\gamma}^{\mathrm{attn}}]_{\alpha:}) = \gamma_\alpha^{\mathrm{attn}} x_\alpha^{\mathrm{norm}} + \beta_\alpha^{\mathrm{attn}}$.

Now, the processed feature $\widetilde{\boldsymbol{x}}^{\mathrm{norm}}$ is put to the multi-head attention layer, which is known to be representable by the NE⊗OR⊤ Program (e.g., see Program 10 in [67]). We write it for completeness here. We denote $\boldsymbol{W}^K, \boldsymbol{W}^Q, \boldsymbol{W}^V \in \mathbb{R}^{n \times n}$ the attention layer's key, query, and value matrix, each row representing a head with dimension 1. The forward pass can be written as follows.

$$\boldsymbol{k} := \boldsymbol{W}^K \widetilde{\boldsymbol{x}}^{\mathrm{norm}}, \quad\quad\quad (\texttt{MatMul})$$
$$\boldsymbol{q} := \boldsymbol{W}^Q \widetilde{\boldsymbol{x}}^{\mathrm{norm}}, \quad\quad\quad (\texttt{MatMul})$$
$$\boldsymbol{v} := \boldsymbol{W}^V \widetilde{\boldsymbol{x}}^{\mathrm{norm}}, \quad\quad\quad (\texttt{MatMul})$$
$$x_\alpha^{\mathrm{attn}} := \psi([\boldsymbol{k}, \boldsymbol{q}, \boldsymbol{v}]_{\alpha:}), \quad\quad (\texttt{OuterNonlin})$$

where $\psi([\boldsymbol{k}, \boldsymbol{q}, \boldsymbol{v}]_{\alpha:}) = k_\alpha q_\alpha v_\alpha$.

Before discussing the MLP layer, we still have a residual connection and interaction from the gate value $\boldsymbol{\theta}^{\mathrm{attn}}$, which can be represented by

$$h_\alpha := \psi([\boldsymbol{x}^{\mathrm{attn}}, \boldsymbol{\theta}^{\mathrm{attn}}, \boldsymbol{x}]_{\alpha:}), \quad\quad (\texttt{OuterNonlin})$$

where $\psi([\boldsymbol{x}^{\mathrm{attn}}, \boldsymbol{\theta}^{\mathrm{attn}}, \boldsymbol{x}]_{\alpha:}) = x_\alpha + \theta_\alpha^{\mathrm{attn}} x_\alpha^{\mathrm{attn}}$.

The remaining part is very similar to the above, which includes the layer norm, interaction with the outputs of adaLN, the MLP layer, and the residual connection. We write them in sequence for clarity.

For the layer norm, we can write

$$\mathbb{E}[\boldsymbol{h}] := \frac{1}{n} \sum_{\alpha=1}^{n} h_\alpha, \quad\quad\quad\quad (\texttt{Avg})$$
$$\bar{h}_\alpha^2 := \psi_1(h_\alpha; \mathbb{E}[\boldsymbol{h}]), \quad\quad\quad (\texttt{OuterNonlin})$$
$$\mathbb{V}[\boldsymbol{h}] := \frac{1}{n} \sum_{\alpha=1}^{n} \bar{h}_\alpha^2, \quad\quad\quad\quad (\texttt{Avg})$$
$$h_\alpha^{\mathrm{norm}} := \psi_2(h_\alpha; [\mathbb{E}[\boldsymbol{h}], \mathbb{V}[\boldsymbol{h}]]), \quad\quad (\texttt{OuterNonlin})$$

where $\psi_1(h_\alpha; \mathbb{E}[\boldsymbol{h}]) = (h_\alpha - \mathbb{E}[\boldsymbol{h}])^2$, $\psi_2(h_\alpha; [\mathbb{E}[\boldsymbol{h}], \mathbb{V}[\boldsymbol{h}]]) = (h_\alpha - \mathbb{E}[\boldsymbol{h}])/\sqrt{\mathbb{V}[\boldsymbol{h}] + \epsilon}$.

For the interaction with scale and shift value from adaLN, we have

$$\widetilde{h}_\alpha^{\mathrm{norm}} := \psi([\boldsymbol{h}^{\mathrm{norm}}, \boldsymbol{\beta}^{\mathrm{mlp}}, \boldsymbol{\gamma}^{\mathrm{mlp}}]_{\alpha:}), \quad (\texttt{OuterNonlin})$$

where $\psi([\boldsymbol{h}^{\mathrm{norm}}, \boldsymbol{\beta}^{\mathrm{mlp}}, \boldsymbol{\gamma}^{\mathrm{mlp}}]_{\alpha:}) := \gamma_\alpha^{\mathrm{mlp}} h_\alpha^{\mathrm{norm}} + \beta_\alpha^{\mathrm{mlp}}$.

For the two-layer MLP in the Transformer block, we use $\boldsymbol{W}^{(1)}, \boldsymbol{W}^{(2)} \in \mathbb{R}^{n \times n}$ to denote the weights in its first layer and the second layer, respectively. We have

$$\boldsymbol{h}_\alpha^{(1)} := \boldsymbol{W}^{(1)} \widetilde{\boldsymbol{h}}^{\mathrm{norm}}, \quad\quad\quad (\texttt{MatMul})$$
$$\widetilde{h}_\alpha^{(1)} := \psi(h_\alpha^{(1)}), \quad\quad\quad\quad (\texttt{OuterNonlin})$$
$$\boldsymbol{h}^{\mathrm{mlp}} := \boldsymbol{W}^{(2)} \widetilde{\boldsymbol{h}}^{(1)}, \quad\quad\quad (\texttt{MatMul})$$

where the $\psi$ is the GeLU activation function.

For the final residual connection, it can be represented by

$$h_\alpha^{\mathrm{Tblock}} := \psi([\boldsymbol{h}^{\mathrm{mlp}}, \boldsymbol{\theta}^{\mathrm{mlp}}, \boldsymbol{h}]_{\alpha:}), \quad\quad (\texttt{OuterNonlin})$$

where $\psi([\boldsymbol{h}^{\mathrm{mlp}}, \boldsymbol{\theta}^{\mathrm{mlp}}, \boldsymbol{h}]_{\alpha:}) = h_\alpha + \theta_\alpha^{\mathrm{mlp}} h_\alpha^{\mathrm{mlp}}$.

**Final layer with adaLN.** The final layer includes the adaLN, layer norm, and a linear projection. We discuss them in sequence.

The adaLN block has parameters $(\boldsymbol{W}^{\beta-\mathrm{final}}, \boldsymbol{W}^{\gamma-\mathrm{final}})$ and receive the condition $\boldsymbol{c}$. Its forward pass can be written as follows.

$$\widetilde{c}_\alpha := \psi(c_\alpha), \qquad\qquad\qquad (\texttt{OuterNonlin})$$

$$\boldsymbol{\beta}^{\mathrm{final}} := \boldsymbol{W}^{\beta-\mathrm{final}}\widetilde{\boldsymbol{c}}, \qquad\qquad\qquad (\texttt{MatMul})$$

$$\boldsymbol{\gamma}^{\mathrm{final}} := \boldsymbol{W}^{\gamma-\mathrm{final}}\widetilde{\boldsymbol{c}}, \qquad\qquad\qquad (\texttt{MatMul})$$

where $\psi$ is the SiLU activation.

The layer norm layer normalizes the output of the transformer block $\boldsymbol{h}^{\mathrm{Tblock}}$, which is the same as the above layer norm. We omit the derivation here for simplicity. We denote the output of the layer norm by $\boldsymbol{z}^{\mathrm{norm}} \in \mathbb{R}^n$ and the parameter of the linear projection by $\boldsymbol{w}^{\mathrm{final}}$, then the remains calculation can be derived as

$$\widetilde{z}_\alpha^{\mathrm{norm}} := \psi_1([\boldsymbol{z}^{\mathrm{norm}}, \boldsymbol{\beta}^{\mathrm{final}}, \boldsymbol{\gamma}^{\mathrm{final}}]_{\alpha:}), \qquad\qquad (\texttt{OuterNonlin})$$

$$\widetilde{z}_\alpha^{\mathrm{final}} := \psi_2([\boldsymbol{w}^{\mathrm{final}}, \widetilde{\boldsymbol{z}}^{\mathrm{norm}}]_{\alpha:}), \qquad\qquad (\texttt{OuterNonlin})$$

$$z^{\mathrm{final}} = \frac{1}{n}\sum_{\alpha=1}^n \widetilde{z}_\alpha^{\mathrm{final}}, \qquad\qquad\qquad (\texttt{Avg})$$

where $\psi_1([\boldsymbol{z}^{\mathrm{norm}}, \boldsymbol{\beta}^{\mathrm{final}}, \boldsymbol{\gamma}^{\mathrm{final}}]_{\alpha:}) := \gamma_\alpha^{\mathrm{final}} z_\alpha^{\mathrm{norm}} + \beta_\alpha^{\mathrm{final}}$ and $\psi_2([\boldsymbol{w}^{\mathrm{final}}, \widetilde{\boldsymbol{z}}^{\mathrm{norm}}]_{\alpha:}) = w_\alpha^{\mathrm{final}} \widetilde{z}_\alpha^{\mathrm{norm}}$. Therefore, the forward pass from the inputs $(x, y, t)$ to output $z^{\mathrm{final}}$ can be represented by the $\mathrm{NE}{\otimes}\mathrm{OR}{\top}$ Program, which means that The $\mu$P of DiT is the same as the standard $\mu$P in Table 6. The proof is completed.

$\square$

## C.2    Proof of PixArt-$\alpha$

*Proof.* Since most parts of the PixArt-$\alpha$ are the same as those in DiT, we only discuss the different modules from DiT for simplicity. These modules are mainly induced from the condition process.

**Embedder for the diffusion timestep $t$.** The embedder for the diffusion timestep $t$ not only outputs $\boldsymbol{t}^{\mathrm{embed}}$ but also gate, shift, and scale values based on $\boldsymbol{t}^{\mathrm{embed}}$ (named adaLN-single in [8]), which are shared by the following Transformer block. We denote the additional parameters by $(\boldsymbol{W}^{\theta-\mathrm{tattn}}, \boldsymbol{W}^{\beta-\mathrm{tattn}}, \boldsymbol{W}^{\gamma-\mathrm{tattn}}, \boldsymbol{W}^{\theta-\mathrm{tmlp}}, \boldsymbol{W}^{\beta-\mathrm{tmlp}}, \boldsymbol{W}^{\gamma-\mathrm{tmlp}})$, each of them is in $\mathbb{R}^{n\times n}$. Then its forward pass can be written as

$$\widetilde{t}_\alpha := \psi(t_\alpha^{\mathrm{embed}}), \qquad\qquad\qquad (\texttt{OuterNonlin})$$

$$\boldsymbol{\theta}^{\mathrm{tattn}} := \boldsymbol{W}^{\theta-\mathrm{tattn}}\widetilde{\boldsymbol{t}}, \qquad\qquad\qquad (\texttt{MatMul})$$

$$\boldsymbol{\beta}^{\mathrm{tattn}} := \boldsymbol{W}^{\beta-\mathrm{tattn}}\widetilde{\boldsymbol{t}}, \qquad\qquad\qquad (\texttt{MatMul})$$

$$\boldsymbol{\gamma}^{\mathrm{tattn}} := \boldsymbol{W}^{\gamma-\mathrm{tattn}}\widetilde{\boldsymbol{t}}, \qquad\qquad\qquad (\texttt{MatMul})$$

$$\boldsymbol{\theta}^{\mathrm{tmlp}} := \boldsymbol{W}^{\theta-\mathrm{tmlp}}\widetilde{\boldsymbol{t}}, \qquad\qquad\qquad (\texttt{MatMul})$$

$$\boldsymbol{\beta}^{\mathrm{tmlp}} := \boldsymbol{W}^{\beta-\mathrm{tmlp}}\widetilde{\boldsymbol{t}}, \qquad\qquad\qquad (\texttt{MatMul})$$

$$\boldsymbol{\gamma}^{\mathrm{tmlp}} := \boldsymbol{W}^{\gamma-\mathrm{tmlp}}\widetilde{\boldsymbol{t}}, \qquad\qquad\qquad (\texttt{MatMul})$$

where $\psi$ is the SiLU activation.

**Transformer block with gate-shift-scale table.**   PixArt-$\alpha$ replaces the adaLN block in DiT block with the gate-shift-scale table, which parameters are denoted by $\boldsymbol{\theta}^{\mathrm{xattn}}, \boldsymbol{\beta}^{\mathrm{xattn}}, \boldsymbol{\gamma}^{\mathrm{xattn}}, \boldsymbol{\theta}^{\mathrm{xmlp}}, \boldsymbol{\beta}^{\mathrm{xmlp}}, \boldsymbol{\gamma}^{\mathrm{xmlp}} \in \mathbb{R}^n$.   The gate-shift-scale table interacts with the shared embeddings of $t$ in the following way.

$$\theta_\alpha^{\mathrm{attn}} := \psi([\boldsymbol{\theta}^{\mathrm{xattn}}, \boldsymbol{\theta}^{\mathrm{tattn}}]_{\alpha:}), \qquad\qquad (\texttt{OuterNonlin})$$

$$\beta_\alpha^{\mathrm{attn}} := \psi([\boldsymbol{\beta}^{\mathrm{xattn}}, \boldsymbol{\beta}^{\mathrm{tattn}}]_{\alpha:}), \qquad\qquad (\texttt{OuterNonlin})$$

$$\gamma_\alpha^{\text{attn}} := \psi([\boldsymbol{\gamma}^{\text{xattn}}, \boldsymbol{\gamma}^{\text{tattn}}]_{\alpha:}), \qquad\qquad (\texttt{OuterNonlin})$$

$$\theta_\alpha^{\text{mlp}} := \psi([\boldsymbol{\theta}^{\text{xmlp}}, \boldsymbol{\theta}^{\text{tmlp}}]_{\alpha:}), \qquad\qquad (\texttt{OuterNonlin})$$

$$\beta_\alpha^{\text{mlp}} := \psi([\boldsymbol{\beta}^{\text{xmlp}}, \boldsymbol{\beta}^{\text{tmlp}}]_{\alpha:}), \qquad\qquad (\texttt{OuterNonlin})$$

$$\gamma_\alpha^{\text{mlp}} := \psi([\boldsymbol{\gamma}^{\text{xmlp}}, \boldsymbol{\gamma}^{\text{tmlp}}]_{\alpha:}), \qquad\qquad (\texttt{OuterNonlin})$$

where $\psi(a, b) = a + b$. The above outputted gate, shift, and scale values are used in the following attention layer and MLP layer, just like DiT.

One more difference between the Transformer block of PixArt-$\alpha$ and DiT is the existence of the additional cross-attention layer, which integrates the output of the self-attention layer $\boldsymbol{h}$ and the text embedding $\boldsymbol{y}^{\text{embed}}$. We denote its parameters by $\boldsymbol{W}^{cK}, \boldsymbol{W}^{cQ}, \boldsymbol{W}^{cV}, \boldsymbol{W}^{\text{proj}} \in \mathbb{R}^{n \times n}$ It can be written as follows.

$$\boldsymbol{k} := \boldsymbol{W}^{cK} \boldsymbol{y}^{\text{embed}}, \qquad\qquad (\texttt{MatMul})$$

$$\boldsymbol{q} := \boldsymbol{W}^{cQ} \boldsymbol{h}, \qquad\qquad (\texttt{MatMul})$$

$$\boldsymbol{v} := \boldsymbol{W}^{cV} \boldsymbol{y}^{\text{embed}}, \qquad\qquad (\texttt{MatMul})$$

$$h_\alpha^v := \psi([\boldsymbol{k}, \boldsymbol{q}, \boldsymbol{v}]_{\alpha:}), \qquad\qquad (\texttt{OuterNonlin})$$

$$\boldsymbol{h}^{\text{cross}} := \boldsymbol{W}^{\text{proj}} \boldsymbol{h}^v, \qquad\qquad (\texttt{MatMul})$$

where $\psi([\boldsymbol{k}, \boldsymbol{q}, \boldsymbol{v}]_{\alpha:}) = k_\alpha q_\alpha v_\alpha$. The following pre-LN MLP with gate-shift-scale table is a trivial extension of the pre-LN MLP layer in the DiT block, like what we did above, so we omit them here.

**Final layer with shift-scale table.** The only difference is that we replace the adaLN block with a shift-scale table, just like what we did for the attention layer, so we omit the derivation here.

$\square$

### C.3 Proof of U-ViT

*Proof.* We only discuss the modules that do not occur in DiT and PixArt-$\alpha$. We find that the only new operator in U-ViT is the long skip connection, which concatenates the current main branch $\boldsymbol{h}^m \in \mathbb{R}^n$ and the long skip branch $\boldsymbol{h}^s \in \mathbb{R}^n$, and then performs a linear projection to reduce the dimension. We denote the parameters of the linear layer by $\boldsymbol{W} = [\boldsymbol{W}^m, \boldsymbol{W}^s]$, where $\boldsymbol{W}^m, \boldsymbol{W}^s \in \mathbb{R}^{n \times n}$, then the long skip connection can be represented by

$$\widetilde{\boldsymbol{h}}^m = \boldsymbol{W}^m \boldsymbol{h}^m, \qquad\qquad (\texttt{MatMul})$$

$$\widetilde{\boldsymbol{h}}^s = \boldsymbol{W}^s \boldsymbol{h}^s, \qquad\qquad (\texttt{MatMul})$$

$$h_\alpha = \psi([\widetilde{\boldsymbol{h}}^m, \widetilde{\boldsymbol{h}}^s]_{\alpha:}), \qquad\qquad (\texttt{OuterNonlin})$$

where $\psi([\widetilde{\boldsymbol{h}}^m, \widetilde{\boldsymbol{h}}^s]_{\alpha:}) = \widetilde{h}_\alpha^m + \widetilde{h}_\alpha^s$. Because other operators in U-ViT (CNN embeddings/decoder, layer norm, attention layer, MLP layer, residual connection) have been studied in DiT, we finish the proof. $\square$

### C.4 Proof of MMDiT

*Proof.* For simplicity, we only discuss the modules that do not occur in the above models. The new module that emerges in the MMDiT is the joint attention with the QK-Norm for the latent feature $\boldsymbol{x}$ and text condition $\boldsymbol{c}$. For the latent feature, we denote the key/query/value parameters by $\boldsymbol{W}^{Kx}, \boldsymbol{W}^{Qx}, \boldsymbol{W}^{Vx} \in \mathbb{R}^{n \times n}$, the parameters of QK-norm by $\boldsymbol{\gamma}^{Kx}, \boldsymbol{\gamma}^{Qx} \in \mathbb{R}^n$, and the parameters of linear projection by $\boldsymbol{W}^{Ox} \in \mathbb{R}^{n \times n}$. We can define the similar parameters for the text condition $\boldsymbol{c}$ as $\boldsymbol{W}^{Kc}, \boldsymbol{W}^{Qc}, \boldsymbol{W}^{Vc}, \boldsymbol{\gamma}^{Kc}, \boldsymbol{\gamma}^{Qc}, \boldsymbol{W}^{Oc}$. The forward pass of the joint attention layer can be written as follows.

$$\boldsymbol{k}^x := \boldsymbol{W}^{Kx} \boldsymbol{x}, \qquad\qquad (\texttt{MatMul})$$

$$\boldsymbol{q}^x := \boldsymbol{W}^{Qx} \boldsymbol{x}, \qquad\qquad (\texttt{MatMul})$$

$$\boldsymbol{v}^x := \boldsymbol{W}^{Vx} \boldsymbol{x}, \qquad\qquad (\texttt{MatMul})$$

$$\boldsymbol{k}^c := \boldsymbol{W}^{Kc} \boldsymbol{c}, \qquad\qquad (\texttt{MatMul})$$

$$\boldsymbol{q}^c := \boldsymbol{W}^{Qc}\boldsymbol{c}, \hspace{4cm} \text{(MatMul)}$$

$$\boldsymbol{v}^c := \boldsymbol{W}^{Vc}\boldsymbol{c}, \hspace{4cm} \text{(MatMul)}$$

$$k_\alpha^{x\text{norm}} := \psi_1([\boldsymbol{k}^x, \boldsymbol{\gamma}^{Kx}]_{\alpha:}), \hspace{3cm} \text{(OuterNonlin)}$$

$$q_\alpha^{x\text{norm}} := \psi_1([\boldsymbol{q}^x, \boldsymbol{\gamma}^{Qx}]_{\alpha:}), \hspace{3cm} \text{(OuterNonlin)}$$

$$k_\alpha^{c\text{norm}} := \psi_1([\boldsymbol{k}^c, \boldsymbol{\gamma}^{Kc}]_{\alpha:}), \hspace{3cm} \text{(OuterNonlin)}$$

$$q_\alpha^{c\text{norm}} := \psi_1([\boldsymbol{q}^c, \boldsymbol{\gamma}^{Qc}]_{\alpha:}), \hspace{3cm} \text{(OuterNonlin)}$$

$$\widetilde{\boldsymbol{v}}^x := \psi_2([\boldsymbol{q}^{x\text{norm}}, \boldsymbol{k}^{x\text{norm}}, \boldsymbol{v}^x, \boldsymbol{k}^{c\text{norm}}, \boldsymbol{v}^c]_{\alpha:}), \hspace{1cm} \text{(OuterNonlin)}$$

$$\widetilde{\boldsymbol{v}}^c := \psi_2([\boldsymbol{q}^{c\text{norm}}, \boldsymbol{k}^{x\text{norm}}, \boldsymbol{v}^x, \boldsymbol{k}^{c\text{norm}}, \boldsymbol{v}^c]_{\alpha:}), \hspace{1cm} \text{(OuterNonlin)}$$

$$\boldsymbol{v}^{x\text{out}} := \boldsymbol{W}^{Ox}\widetilde{\boldsymbol{v}}^x, \hspace{4cm} \text{(MatMul)}$$

$$\boldsymbol{v}^{c\text{out}} := \boldsymbol{W}^{Oc}\widetilde{\boldsymbol{v}}^c, \hspace{4cm} \text{(MatMul)}$$

where $\psi_1(a, b) = ab/|a|$ and $\psi_2(a, b, c, d, e) = abc + ade$. The proof is completed. $\hspace{1cm}\square$

## Appendix D   Additional Details for Section 3

In this section, we summarize the methodology for verifying base HP transferability across widths, batch sizes, and training steps in Algorithms 1, 2, and 3, respectively. We also present the $\mu$Transfer algorithm [73] in Algorithm 4 for the completeness.

---

**Algorithm 1** Validate the base HP Transferability of $\mu$P across widths

---

1: **Input:** set of widths $\{n_i\}_{i=1}^P$, set of base HP to validate $\{\lambda_j\}_{j=1}^R$, other fixed base HPs $\boldsymbol{\theta}$, batch size $B$, training iteration $T$, base width $n_{\text{base}}$
2: **Output:** whether model with $\mu$P enjoy base HP transferability across widths
3: **for** $i = 1$ **to** $P$ **do**
4:     **for** $j = 1$ **to** $R$ **do**
5:         $\lambda_{ij} \leftarrow \mu\text{P}(\lambda_j, n_{\text{base}}, n_i)$                   $\triangleright$ actual HP for $n_i$-width model
6:         $\boldsymbol{\theta}_i \leftarrow \mu\text{P}(\boldsymbol{\theta}, n_{\text{base}}, n_i)$                   $\triangleright$ actual HP for $n_i$-width model
7:         $\mathcal{M}_{ij} \leftarrow$ train $n_i$-width model with b.s. $B$, iter. $T$, HPs $\lambda_{ij}, \boldsymbol{\theta}_i$
8:         $s_{ij} \leftarrow$ evaluate score for $\mathcal{M}_{ij}$
9:     **end for**
10:    $\lambda_i^* \leftarrow \arg\max_{\lambda_j}(\{s_{ij}\}_{j=1}^R)$         $\triangleright$ Obtain the optimal base HP for $n_i$-width model
11: **end for**
12: **return** whether $\lambda_i^*$s are approximately same

---

---

**Algorithm 2** Validate the base HP Transferability of $\mu$P across batch sizes

---

1: **Input:** set of batch sizes $\{B_i\}_{i=1}^P$, set of base HP to validate $\{\lambda_j\}_{j=1}^R$, other fixed base HPs $\boldsymbol{\theta}$, width $n$, training iteration $T$, base width $n_{\text{base}}$
2: **Output:** whether model with $\mu$P enjoy base HP transferability across batch sizes
3: **for** $i = 1$ **to** $P$ **do**
4:     **for** $j = 1$ **to** $R$ **do**
5:         $\lambda_{ij} \leftarrow \mu\text{P}(\lambda_j, n_{\text{base}}, n)$                   $\triangleright$ actual HP for $n$-width model
6:         $\boldsymbol{\theta}_i \leftarrow \mu\text{P}(\boldsymbol{\theta}, n_{\text{base}}, n)$                   $\triangleright$ actual HP for $n$-width model
7:         $\mathcal{M}_{ij} \leftarrow$ train $n$-width model with b.s. $B_i$, iter. $T$, HPs $\lambda_{ij}, \boldsymbol{\theta}_i$
8:         $s_{ij} \leftarrow$ evaluate score for $\mathcal{M}_{ij}$
9:     **end for**
10:    $\lambda_i^* \leftarrow \arg\max_{\lambda_j}(\{s_{ij}\}_{j=1}^R)$         $\triangleright$ Obtain the optimal base HP for $n$-width model
11: **end for**
12: **return** whether $\lambda_i^*$s are approximately same

---

---

**Algorithm 3** Validate the base HP Transferability of $\mu$P across training steps

---

1: **Input:** set of training steps $\{T_i\}_{i=1}^P$, set of base HP to validate $\{\lambda_j\}_{j=1}^R$, other fixed base HPs $\boldsymbol{\theta}$, batch size $B$, width $n$, base width $n_{\text{base}}$
2: **Output:** whether model with $\mu$P enjoy base HP transferability across training steps
3: **for** $i = 1$ **to** $P$ **do**
4:     **for** $j = 1$ **to** $R$ **do**
5:         $\lambda_{ij} \leftarrow \mu\text{P}(\lambda_j, n_{\text{base}}, n)$                             $\triangleright$ actual HP for $n$-width model
6:         $\boldsymbol{\theta}_i \leftarrow \mu\text{P}(\boldsymbol{\theta}, n_{\text{base}}, n)$                             $\triangleright$ actual HP for $n$-width model
7:         $\mathcal{M}_{ij} \leftarrow$ train $n$-width model with b.s. $B$, iter. $T_i$, HPs $\lambda_{ij}, \boldsymbol{\theta}_i$
8:         $s_{ij} \leftarrow$ evaluate score for $\mathcal{M}_{ij}$
9:     **end for**
10:     $\lambda_i^* \leftarrow \arg\max_{\lambda_j}(\{s_{ij}\}_{j=1}^R)$             $\triangleright$ Obtain the optimal base HP for $n$-width model
11: **end for**
12: **return** whether $\lambda_i^*$s are approximately same

---

---

**Algorithm 4** $\mu$Transfer (Algorithm 1 in [73])

---

1: **Input:** base/proxy/target width $n_{\text{base}}/n_{\text{proxy}}/n_{\text{target}}$, proxy/target batch size $B_{\text{proxy}}/B_{\text{target}}$, proxy/target training steps $T_{\text{proxy}}/T_{\text{target}}$, set of base HPs to search $\{\boldsymbol{\theta}_i\}_{j=1}^R$
2: **Output:** Trained target model $\mathcal{M}_{\text{target}}$
3: **for** $j = 1$ **to** $R$ **do**
4:     $\boldsymbol{\theta}_{\text{proxy},j} \leftarrow \mu\text{P}(\boldsymbol{\theta}_j, n_{\text{base}}, n_{\text{proxy}})$              $\triangleright$ actual HPs of proxy model
5:     $\mathcal{M}_j \leftarrow$ train $n_{\text{proxy}}$-width model with batch size $B_{\text{proxy}}$, training steps $T_{\text{proxy}}$, HPs $\boldsymbol{\theta}_{\text{proxy},j}$
6:     $s_j \leftarrow$ evaluate score for $\mathcal{M}_j$
7: **end for**
8: $\boldsymbol{\theta}^* \leftarrow \arg\max_{\boldsymbol{\theta}_j}(\{s_j\}_{j=1}^R)$
9: $\boldsymbol{\theta}_{\text{target}} \leftarrow \mu\text{P}(\boldsymbol{\theta}^*, n_{\text{base}}, n_{\text{target}})$                  $\triangleright$ actual HPs of target model
10: $\mathcal{M}_{\text{target}} \leftarrow$ train $n_{\text{target}}$-width model with batch size $B_{\text{target}}$, training steps $T_{\text{target}}$, HPs $\boldsymbol{\theta}_{\text{target}}$
11: **return** $\mathcal{M}_{\text{target}}$

---

## Appendix E    Additional Experimental Details and Results

In this section, we present the additional experimental details and results which are neglected by the main text.

### E.1    Assets and Licenses

All used assets (datasets and codes) and their licenses are listed in Table 7.

Table 7: **The used assets and licenses.**

| URL | Citation | License |
|---|---|---|
| https://www.image-net.org/ | [13] | non-commercial |
| https://ai.meta.com/datasets/segment-anything/ | [33] | Apache-2.0 |
| http://images.cocodataset.org/zips/val2014.zip | [45] | Commons Attribution 4.0 |
| https://huggingface.co/datasets/playgroundai/MJHQ-30K | [39] | - |
| https://github.com/djghosh13/geneval | [21] | MIT |
| https://github.com/facebookresearch/DiT | [52] | Link |
| https://github.com/openai/guided-diffusion | [15] | MIT |
| https://github.com/PixArt-alpha/PixArt-alpha | [8] | Apache-2.0 |
| https://github.com/microsoft/mup | [73] | MIT |

### E.2 Additional Details of DiT Experiments

#### E.2.1 Evaluation Implementation

Following the Table 4 in original DiT paper [52], we use the codebase of ADM [15] to obtain the results of FID, sFID, IS, precision, and recall without cfg.

#### E.2.2 Additional Results of Base HP transferability

**Base learning rate.** Comprehensive results of base learning rate transferability across widths, batch sizes, and training steps are presented in Tables 8, 9, and 10, respectively.

Table 8: **DiT-$\mu$P enjoys base learning rate transferability across widths.** We use a batch size of 256 and 200K training iterations. NaN data points indicate training instability, where the loss explodes. Under $\mu$P, the base learning rate can be transferred across model widths.

| Width | $\log_2(\text{lr})$ | FID-50K $\downarrow$ | sFID-50K $\downarrow$ | Inception Score $\uparrow$ | Precision $\uparrow$ | Recall $\uparrow$ |
|---|---|---|---|---|---|---|
| 144 | -9 | NaN | NaN | NaN | NaN | NaN |
| 144 | -10 | **88.89** | **16.19** | **14.24** | **0.30** | **0.437** |
| 144 | -11 | 91.88 | 16.79 | 13.73 | 0.29 | 0.431 |
| 144 | -12 | 93.61 | 17.94 | 13.41 | 0.28 | 0.406 |
| 144 | -13 | 102.99 | 21.26 | 11.93 | 0.25 | 0.366 |
| 288 | -9 | NaN | NaN | NaN | NaN | NaN |
| 288 | -10 | **61.65** | **9.06** | **20.60** | **0.41** | **0.563** |
| 288 | -11 | 63.85 | 10.32 | 20.59 | 0.40 | 0.561 |
| 288 | -12 | 65.99 | 10.79 | 19.73 | 0.38 | 0.544 |
| 288 | -13 | 79.17 | 13.04 | 15.92 | 0.33 | 0.503 |
| 576 | -9 | NaN | NaN | NaN | NaN | NaN |
| 576 | -10 | **43.73** | **6.65** | **28.82** | **0.51** | **0.611** |
| 576 | -11 | 45.00 | 7.22 | 28.61 | 0.50 | 0.606 |
| 576 | -12 | 50.30 | 7.87 | 25.67 | 0.47 | 0.602 |
| 576 | -13 | 66.20 | 9.67 | 18.91 | 0.39 | 0.562 |

**Base output multiplier.** Likewise that for the base learning rate, we also sweep the base multiplier of the output weight over the set $\{2^{-6}, 2^{-4}, 2^{-2}, 2^0, 2^2, 2^4, 2^6\}$ across various widths, batch sizes and training steps. FID-50K results are shown in Figure 7, and comprehensive results for other metrics are presented in Tables 11, 12, and 13, respectively. As presented in Figure 7, the optimal base output multiplier is approximately transferable across different widths, batch sizes, and training iterations.

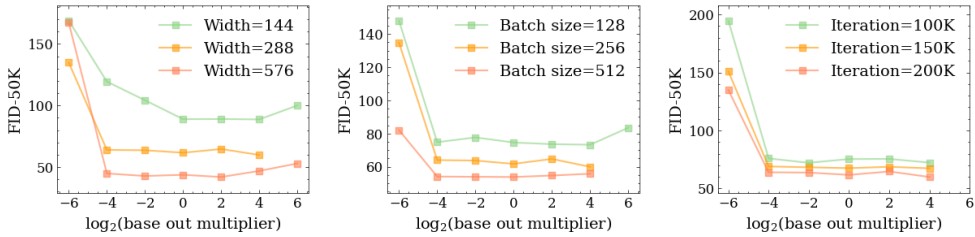

(a) HP transfer across widths.  (b) HP transfer across batch sizes.  (c) HP transfer across iterations.

Figure 7: **DiT-$\mu$P enjoys base HP transferability**. Unless otherwise specified, we use a model width of 288, a batch size of 256, and a training iteration of 200K. Missing data points indicate training instability. Under $\mu$P, the base output multiplier can be transferred across model widths, batch sizes, and steps.

Table 9: **DiT-$\mu$P enjoys base learning rate transferability across batch sizes.** We use a width of 288 and a training iteration of 200K. NaN data points indicate training instability, where the loss explodes. We **bold** the best result and underline the second-best result. Under $\mu$P, the base learning rate can be (approximately) transferred across batch sizes.

| Batch size | $\log_2(\mathrm{lr})$ | FID-50K ↓ | sFID-50K ↓ | Inception Score ↑ | Precision ↑ | Recall ↑ |
|---|---|---|---|---|---|---|
| 128 | -9 | NaN | NaN | NaN | NaN | NaN |
| 128 | -10 | 74.54 | **10.32** | 16.70 | 0.358 | 0.513 |
| 128 | -11 | **73.47** | 10.33 | **16.83** | **0.366** | **0.523** |
| 128 | -12 | 76.98 | 12.12 | 16.50 | 0.343 | 0.498 |
| 128 | -13 | 88.79 | 14.63 | 14.10 | 0.293 | 0.453 |
| 256 | -9 | NaN | NaN | NaN | NaN | NaN |
| 256 | -10 | **61.65** | **9.06** | **20.60** | **0.407** | **0.563** |
| 256 | -11 | 63.85 | 10.32 | 20.59 | 0.398 | 0.561 |
| 256 | -12 | 65.99 | 10.79 | 19.73 | 0.383 | 0.544 |
| 256 | -13 | 79.17 | 13.04 | 15.92 | 0.327 | 0.503 |
| 512 | -9 | 53.90 | **8.52** | 24.18 | **0.452** | 0.585 |
| 512 | -10 | **53.72** | 9.27 | 24.66 | 0.448 | **0.592** |
| 512 | -11 | 54.36 | 9.33 | **24.96** | 0.442 | 0.587 |
| 512 | -12 | 57.60 | 10.13 | 23.37 | 0.420 | 0.577 |
| 512 | -13 | 72.09 | 12.15 | 17.81 | 0.356 | 0.537 |

Table 10: **DiT-$\mu$P enjoys base learning rate transferability across training steps.** We use a width of 288 and a batch size of 256. NaN data points indicate training instability, where the loss explodes. Under $\mu$P, the base learning rate can be transferred across training steps.

| Iteration | $\log_2(\mathrm{lr})$ | FID-50K ↓ | sFID-50K ↓ | Inception Score ↑ | Precision ↑ | Recall ↑ |
|---|---|---|---|---|---|---|
| 100K | -9 | **74.40** | 11.37 | **16.75** | 0.363 | 0.504 |
| 100K | -10 | 75.35 | **10.67** | 16.61 | **0.363** | **0.514** |
| 100K | -11 | 78.57 | 11.69 | 16.17 | 0.338 | 0.495 |
| 100K | -12 | 84.74 | 12.97 | 14.78 | 0.315 | 0.474 |
| 100K | -13 | 102.79 | 18.37 | 11.74 | 0.243 | 0.363 |
| 150K | -9 | 70.44 | 10.07 | 18.03 | **0.396** | 0.534 |
| 150K | -10 | **67.48** | **9.45** | **18.83** | 0.387 | **0.553** |
| 150K | -11 | 69.61 | 10.51 | 18.45 | 0.380 | 0.533 |
| 150K | -12 | 72.72 | 11.33 | 17.52 | 0.357 | 0.515 |
| 150K | -13 | 88.05 | 14.07 | 14.34 | 0.296 | 0.455 |
| 200K | -9 | NaN | NaN | NaN | NaN | NaN |
| 200K | -10 | **61.65** | **9.06** | **20.60** | **0.407** | **0.563** |
| 200K | -11 | 63.85 | 10.32 | 20.59 | 0.398 | 0.561 |
| 200K | -12 | 65.99 | 10.79 | 19.73 | 0.383 | 0.544 |
| 200K | -13 | 79.17 | 13.04 | 15.92 | 0.327 | 0.503 |

Table 11: **DiT-$\mu$P enjoys base output multiplier transferability across widths.** We use a batch size of 256 and a training iteration of 200K. NaN data points indicate training instability, where the loss explodes. Under $\mu$P, the base output multiplier can be (approximately) transferred across model widths.

| Width | $\log_2(\phi_{\text{out}})$ | FID-50K $\downarrow$ | sFID-50K $\downarrow$ | Inception Score $\uparrow$ | Precision $\uparrow$ | Recall $\uparrow$ |
|---|---|---|---|---|---|---|
| 144 | -6 | 168.81 | 62.70 | 6.30 | 0.157 | 0.061 |
| 144 | -4 | 119.38 | 27.05 | 10.77 | 0.185 | 0.300 |
| 144 | -2 | 104.26 | 15.65 | 12.17 | 0.262 | 0.419 |
| 144 | 0 | 88.89 | 16.19 | 14.24 | 0.297 | 0.437 |
| 144 | 2 | 89.01 | 16.83 | 13.97 | 0.294 | 0.433 |
| 144 | 4 | 88.68 | 15.71 | 14.19 | 0.303 | 0.441 |
| 144 | 6 | 100.03 | 19.33 | 12.48 | 0.262 | 0.386 |
| 288 | -6 | 134.82 | 41.52 | 9.44 | 0.184 | 0.186 |
| 288 | -4 | 63.93 | 8.98 | 20.31 | 0.395 | 0.552 |
| 288 | -2 | 63.65 | 9.15 | 20.23 | 0.401 | 0.563 |
| 288 | 0 | 61.65 | 9.06 | 20.60 | 0.407 | 0.563 |
| 288 | 2 | 64.64 | 9.44 | 19.30 | 0.398 | 0.555 |
| 288 | 4 | 59.85 | 9.39 | 20.95 | 0.415 | 0.566 |
| 288 | 6 | NaN | NaN | NaN | NaN | NaN |
| 576 | -6 | 167.14 | 69.32 | 5.04 | 0.130 | 0.051 |
| 576 | -4 | 44.82 | 6.66 | 29.47 | 0.510 | 0.592 |
| 576 | -2 | 42.75 | 6.68 | 30.73 | 0.526 | 0.600 |
| 576 | 0 | 43.73 | 6.65 | 28.82 | 0.509 | 0.611 |
| 576 | 2 | 41.93 | 6.68 | 29.45 | 0.537 | 0.600 |
| 576 | 4 | 46.87 | 6.74 | 26.78 | 0.501 | 0.594 |
| 576 | 6 | 52.93 | 7.30 | 24.07 | 0.455 | 0.584 |

Table 12: **DiT-$\mu$P enjoys base output multiplier transferability across batch sizes.** We use a width of 288 and a training iteration of 200K. NaN data points indicate training instability, where the loss explodes. Under $\mu$P, the base output multiplier can be (approximately) transferred across batch sizes.

| Batch Size | $\log_2(\phi_{\text{out}})$ | FID-50K $\downarrow$ | sFID-50K $\downarrow$ | Inception Score $\uparrow$ | Precision $\uparrow$ | Recall $\uparrow$ |
|---|---|---|---|---|---|---|
| 128 | -6 | 148.06 | 51.95 | 7.76 | 0.170 | 0.126 |
| 128 | -4 | 74.74 | 10.21 | 17.23 | 0.357 | 0.513 |
| 128 | -2 | 77.62 | 11.55 | 16.20 | 0.346 | 0.511 |
| 128 | 0 | 74.54 | 10.32 | 16.70 | 0.358 | 0.513 |
| 128 | 2 | 73.55 | 10.24 | 16.68 | 0.365 | 0.514 |
| 128 | 4 | 73.19 | 9.93 | 16.93 | 0.375 | 0.514 |
| 128 | 6 | 83.48 | 11.79 | 14.92 | 0.323 | 0.464 |
| 256 | -6 | 134.82 | 41.52 | 9.44 | 0.184 | 0.186 |
| 256 | -4 | 63.93 | 8.98 | 20.31 | 0.395 | 0.552 |
| 256 | -2 | 63.65 | 9.15 | 20.23 | 0.401 | 0.563 |
| 256 | 0 | 61.65 | 9.06 | 20.60 | 0.407 | 0.563 |
| 256 | 2 | 64.64 | 9.44 | 19.30 | 0.398 | 0.555 |
| 256 | 4 | 59.85 | 9.39 | 20.95 | 0.415 | 0.566 |
| 256 | 6 | NaN | NaN | NaN | NaN | NaN |
| 512 | -6 | 82.00 | 14.83 | 16.13 | 0.325 | 0.468 |
| 512 | -4 | 53.99 | 8.83 | 24.49 | 0.447 | 0.585 |
| 512 | -2 | 53.85 | 9.11 | 24.88 | 0.443 | 0.584 |
| 512 | 0 | 53.72 | 9.27 | 24.66 | 0.448 | 0.592 |
| 512 | 2 | 54.62 | 9.14 | 23.97 | 0.449 | 0.593 |
| 512 | 4 | 55.68 | 8.49 | 23.29 | 0.441 | 0.597 |
| 512 | 6 | NaN | NaN | NaN | NaN | NaN |

Table 13: **DiT-$\mu$P enjoys base output multiplier transferability across training steps.** We use a width of 288 and a batch size of 256. NaN data points indicate training instability, where the loss explodes. Under $\mu$P, the base output multiplier can be (approximately) transferred across training steps.

| Iteration | $\log_2(\phi_{\text{out}})$ | FID-50K $\downarrow$ | sFID-50K $\downarrow$ | Inception Score $\uparrow$ | Precision $\uparrow$ | Recall $\uparrow$ |
|---|---|---|---|---|---|---|
| 100K | -6 | 194.59 | 87.88 | 4.38 | 0.084 | 0.016 |
| 100K | -4 | 75.87 | 10.28 | 16.64 | 0.350 | 0.508 |
| 100K | -2 | 72.09 | 9.43 | 17.15 | 0.352 | 0.523 |
| 100K | 0 | 75.35 | 10.67 | 16.61 | 0.363 | 0.514 |
| 100K | 2 | 75.43 | 9.87 | 16.57 | 0.361 | 0.529 |
| 100K | 4 | 72.26 | 9.47 | 17.08 | 0.378 | 0.514 |
| 100K | 6 | NaN | NaN | NaN | NaN | NaN |
| 150K | -6 | 150.86 | 48.06 | 8.17 | 0.163 | 0.143 |
| 150K | -4 | 68.94 | 9.26 | 18.57 | 0.377 | 0.541 |
| 150K | -2 | 68.24 | 9.77 | 18.78 | 0.386 | 0.535 |
| 150K | 0 | 67.48 | 9.45 | 18.83 | 0.387 | 0.553 |
| 150K | 2 | 68.67 | 9.29 | 17.96 | 0.383 | 0.545 |
| 150K | 4 | 67.32 | 9.08 | 18.17 | 0.397 | 0.539 |
| 150K | 6 | NaN | NaN | NaN | NaN | NaN |
| 200K | -6 | 134.82 | 41.52 | 9.44 | 0.184 | 0.186 |
| 200K | -4 | 63.93 | 8.98 | 20.31 | 0.395 | 0.552 |
| 200K | -2 | 63.65 | 9.15 | 20.23 | 0.401 | 0.563 |
| 200K | 0 | 61.65 | 9.06 | 20.60 | 0.407 | 0.563 |
| 200K | 2 | 64.64 | 9.44 | 19.30 | 0.398 | 0.555 |
| 200K | 4 | 59.85 | 9.39 | 20.95 | 0.415 | 0.566 |
| 200K | 6 | NaN | NaN | NaN | NaN | NaN |

### E.2.3 Additional Results of DiT-XL-2 Pretraining

Complete benchmark results of DiT-XL-2 and DiT-XL-2-$\mu$P throughout the training are provided in Table 14. DiT-XL-2-$\mu$P, using a base learning rate transferred from small proxy models, consistently outperforms the original DiT-XL-2 throughout the training process.

Table 14: **Benchmark results of DiT-XL-2 and DiT-XL-2-$\mu$P without classifier-free guidance.** Both models are trained on the ImageNet 256×256 dataset. Results with * are reported in the original paper [52], and others are reproduced by us. DiT-XL-2-$\mu$P, using a base learning rate transferred from small proxy models, consistently outperforms the original baseline throughout the training process.

| Iteration | Method | FID-50K ↓ | sFID-50K ↓ | Inception Score ↑ | Precision ↑ | Recall ↑ |
|---|---|---|---|---|---|---|
| 0.1M | DiT-XL-2 [52] | 48.17 | 7.33 | 26.61 | 0.49 | **0.595** |
| | DiT-XL-2-$\mu$P | **44.15** | **6.72** | **28.54** | **0.54** | 0.575 |
| 0.4M | DiT-XL-2 [52] | 20.38 | 6.37 | 65.02 | 0.63 | **0.635** |
| | DiT-XL-2-$\mu$P | **18.63** | **5.49** | **67.74** | **0.66** | 0.606 |
| 0.8M | DiT-XL-2 [52] | 14.73 | 6.37 | 85.03 | 0.66 | **0.638** |
| | DiT-XL-2-$\mu$P | **12.91** | **5.48** | **89.46** | **0.68** | 0.634 |
| 1.2M | DiT-XL-2 [52] | 12.78 | 6.41 | 95.13 | 0.66 | **0.644** |
| | DiT-XL-2-$\mu$P | **11.25** | **5.56** | **99.89** | **0.68** | 0.638 |
| 1.6M | DiT-XL-2 [52] | 11.79 | 6.49 | 100.88 | 0.67 | **0.651** |
| | DiT-XL-2-$\mu$P | **10.16** | **5.59** | **107.58** | **0.69** | 0.649 |
| 2M | DiT-XL-2 [52] | 10.97 | 6.45 | 105.86 | 0.67 | **0.657** |
| | DiT-XL-2-$\mu$P | **9.72** | **5.62** | **111.23** | **0.69** | 0.651 |
| 2.4M | DiT-XL-2 [52] | 10.75 | 6.59 | 108.23 | 0.67 | **0.665** |
| 7M | DiT-XL-2 [52] | 9.62* | - | - | - | - |
| 2.4M | DiT-XL-2-$\mu$P | **9.44** | **5.66** | **112.98** | **0.68** | 0.653 |

### E.2.4 Computational Cost

It takes 104 (13×8) A100-80GB hours to train the DiT-$\mu$P with a width of 288, a batch size of 256, and a train iteration of 200K steps. The computational cost of other DiT-$\mu$P proxy models can be inferred based on this situation. It takes around 224 (32×7) A100-80GB days to reproduce the pretraining of DiT-XL-2-$\mu$P.

### E.3 Additional Details of PixArt-α Experiments

#### E.3.1 FLOPs ratio of HP Search on Proxy Models

Based on Equation (1) in the main text, we can derive that

$$\texttt{ratio} = \frac{RS_{\text{proxy}}E_{\text{proxy}}}{S_{\text{target}}E_{\text{target}}} = \frac{5 \times 0.04B \times 5}{0.61B \times 30} \approx 5.5\%.$$

Therefore, the HP tuning requires only $5.5\%$ FLOPs of a single training run for PixArt-α.

#### E.3.2 Additional Results of Base Learning Rate Search

We present the detailed GenEval results [21] of different base learning rates in Table 15. Overall, the base learning rate $2^{-10}$ achieves the best GenEval performance.

Table 15: **GenEval results of base learning rate search on PixArt-α-μP proxy models.** 0.04B models with different learning rates are trained for 5 epochs on the SAM dataset. Overall, the base learning rate $2^{-10}$ achieves the best GenEval performance.

| $\log_2(\text{lr})$ | Overall ↑ | Single | Two | Counting | Colors | Position | Color Attribution |
|---|---|---|---|---|---|---|---|
| -9 | | | | | | | |
| -10 | **0.083** | **30.63** | 4.04 | **2.5** | **12.23** | 0.5 | 0 |
| -11 | 0.078 | 29.38 | **5.05** | 1.25 | 9.57 | **1.5** | **0.25** |
| -12 | 0.030 | 12.81 | 0 | 0.31 | 4.79 | 0.25 | 0 |
| -13 | 0.051 | 20.62 | 3.54 | 0.31 | 5.59 | 0.25 | 0 |

#### E.3.3 Additional Results of PixArt-α Pretraining

We present the complete comparison between PixArt-α and PixArt-α-μP throughout training in Table 16. The results demonstrate that PixArt-α-μP consistently outperforms PixArt-α across all evaluation metrics during training.

Table 16: **Benchmark comparison between PixArt-α-μP and PixArt-α.** Both models are trained on the SAM dataset for 30 epochs. PixArt-α-μP with transferred base learning rate from the optimal 0.04B proxy model consistently outperforms the original baseline throughout the training process.

| Epoch | Method | GenEval ↑ | MJHQ | | MS-COCO | |
|---|---|---|---|---|---|---|
| | | | FID-30K ↓ | CLIP Score ↑ | FID-30K ↓ | CLIP Score ↑ |
| 6 | PixArt-α [8] | 0.14 | 43.19 | 25.16 | 42.25 | 27.02 |
| | PixArt-α-μP (Ours) | **0.17** | **34.24** | **25.99** | **32.62** | **28.16** |
| 10 | PixArt-α [8] | 0.19 | 38.36 | 25.78 | 34.58 | 28.12 |
| | PixArt-α-μP (Ours) | **0.20** | **33.35** | **26.25** | **29.68** | **28.87** |
| 16 | PixArt-α [8] | 0.22 | 36.19 | 26.21 | 33.71 | 28.35 |
| | PixArt-α-μP (Ours) | **0.23** | **32.28** | **26.67** | **28.03** | **29.45** |
| 20 | PixArt-α [8] | 0.20 | 35.68 | 26.54 | 30.13 | 28.81 |
| | PixArt-α-μP | **0.23** | **33.42** | **26.83** | **29.05** | **29.53** |
| 26 | PixArt-α [8] | 0.18 | 38.39 | 26.44 | 34.98 | 29.17 |
| | PixArt-α-μP (Ours) | **0.28** | **32.34** | **27.17** | **27.59** | **29.83** |
| 30 | PixArt-α [8] | 0.15 | 42.71 | 26.25 | 37.61 | 28.91 |
| | PixArt-α-μP (Ours) | **0.26** | **29.96** | **27.13** | **25.84** | **29.58** |

### E.4 Additional Details of MMDiT Experiments

#### E.4.1 HPs Tuned by Human Experts

Algorithmic experts take roughly 5 times the cost of full pretraining to tune HPs based on their experience. The best HPs are a learning rate of 2E-4, a gradient clip of 0.1, a REPA loss weight of 0.5 and a warm-up iteration of 1K.

#### E.4.2 Additional Details of Base HP Search

Based on the optimal base HPs searched from 80 trials trained for 30K steps, we further evaluate the HP transferability across iterations. We conducted an additional 5 proxy training runs (100K steps each) using the searched optimal HPs and five different base learning rates, as detailed in Table 17.

Table 17: **Iteration of 30K is enough for proxy task.** The base learning rate of 2.5E-4 is optimal when proxy models are trained for 100K steps, consistent with the results observed at 30K steps.

| Base learning rate | 1E-4 | 2E-4 | 2.5E-4 | 4E-4 | 8E-4 |
|---|---|---|---|---|---|
| Training loss | 0.185028 | 0.184505 | **0.184423** | 0.184436 | 0.185562 |

#### E.4.3 FLOPs ratio of HP Search on Proxy Models

Based on Equation (1) in the main text, we can derive the ratio of tuning cost to single pretraining cost as

$$\texttt{ratio} = \frac{RS_{\text{proxy}}E_{\text{proxy}}}{S_{\text{target}}E_{\text{target}}} = \frac{80 \times 0.18B \times 30K + 5 \times 0.18B \times 100K}{18B \times 200K} = 14.5\%.$$

Besides, as algorithmic experts take roughly 5 times the cost of full pretraining, the ratio of $\mu$P tuning cost to standard human tuning cost is approximately 3%.

#### E.4.4 Additional Results of Training Loss Comparison

We present the complete comparison between the training loss of MMDiT-18B and MMDiT-$\mu$P-18B in Figure 8. MMDiT-$\mu$P-18B achieves consistently lower training loss than the baseline after 15K steps, and the advantage is gradually increasing.

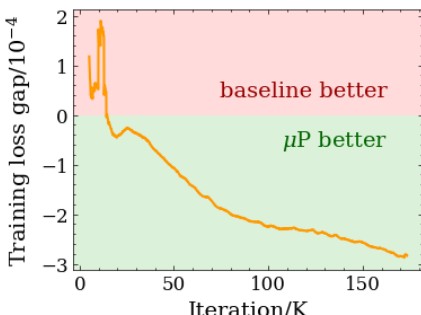

Figure 8: **Comparision between training loss.** We present the training loss of MMDiT-$\mu$P-18B minus that of MMDiT-18B. The training loss gap less than 0 means that MMDiT-$\mu$P-18B is better. MMDiT-$\mu$P-18B achieves consistently lower training loss than the baseline after 15K steps, and the advantage is gradually expanding.

#### E.4.5 Additional Details of Human Evaluation

In this section, we provide the details about the user study in Section 5.2. To evaluate the text-image alignment, we created an internal benchmark comprising 150 prompts, where each prompt includes an average of five binary alignment checks (yes or no), covering a wide range of concepts such as nouns, verbs, adjectives (e.g., size, color, style), and relational terms (e.g., spatial relationships, size comparisons). Ten human experts conducted the evaluation, with each prompt generating three

images, resulting in a total of 22,500 alignment tests. The final score is computed as the average correctness across all test points. We present two examples in Figure 9.

| Prompt | Image | Alignment Evaluation |
|---|---|---|
| One wine glass, two bottles of wine, three cans of beer |  | 1. Wine glass: ✅
2. One (wine glass): ✅
3. Bottles of wine: ✅
4. Two (bottles of wine): ✅
5. Cans of beer: ✅
6. Three (cans of beer): ❌ |
| The Sydney Opera House with the Eiffel tower sitting on the right, and Mount Everest rising above |  | 1. Sydney Opera House: ✅
2. Eiffel tower: ✅
3. Right: ✅
4. Mount Everest: ✅
5. Rising above: ✅ |

Figure 9: **Examples of text-image alignment test evaluated by human.** Ten human experts conducted the evaluation, with each prompt containing an average of five test points and generating three images, resulting in 22,500 alignment test points. The final score is computed as the average correctness across all test points.

### E.4.6 Additional Results of Visualization Comparison

In this section, we provide visual comparisons between MMDiT-$\mu$P-18B and MMDiT-18B baseline in Figure 10.

| Baseline | μP | Baseline | μP |
|---|---|---|---|

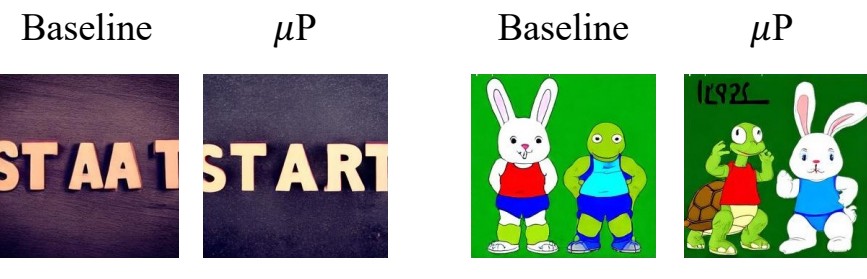

*The word 'START'*     *a white rabbit in blue jogging clothes, a turtle wearing a red tank top*

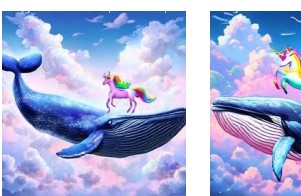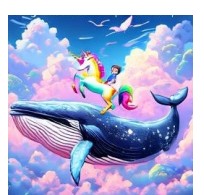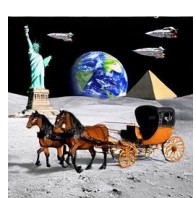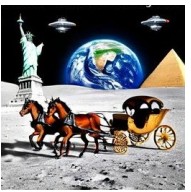

*A huge whale soars through the clouds, with a colorful unicorn standing on its back, and a little girl riding the unicorn*     *Horses are pulling a carriage on the lunar surface. The background is the Statue of Liberty and the Great Pyramid. The Earth and two spaceships can be seen in the sky*

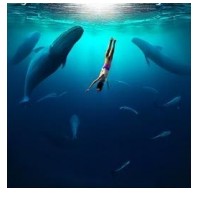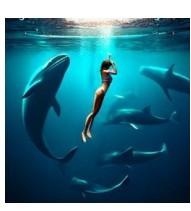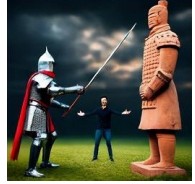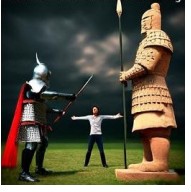

*Seen from above the sea surface, a girl in a swimsuit is diving underwater with a group of whales circling around her, in a long shot, cinematic, realistic style*     *A tall terracotta warrior wielding a spear is confronting a Western knight in metallic armor, both holding their weapons. In between them, a modern person stands with arms outstretched, trying to mediate.*

Figure 10: **Visual comparison between MMDiT-μP-18B and MMDiT-18B baseline.** The configurations for generating images (e.g., random seed, cfg value) of two models are the same. MMDiT-μP-18B shows better text-image alignment than the baseline model.

