# OpenReview forum: "Scaling Diffusion Transformers Efficiently via $\mu$P"
_NeurIPS.cc/2025/Conference — NeurIPS 2025 poster_

### Official Review · Reviewer_mn36 · 2025-06-20

**Clarity:** 2
**Significance:** 2
**Originality:** 2
**Rating:** 3
**Confidence:** 5

**Summary:**

This paper investigates the applicability of Maximal Update Parametrization (µP) to diffusion Transformers, a class of models foundational to vision generative tasks but constrained by costly hyperparameter tuning at scale.

While µP has proven effective for vanilla Transformers in language models, its effectiveness for diffusion Transformers—architecturally and functionally distinct—remained uncertain. The authors extend µP theory to diffusion Transformers such as DiT, U-ViT, PixArt-α, and MMDiT, showing its theoretical alignment with vanilla Transformers.

Empirical results demonstrate that models like DiT-XL-2-µP achieve significantly faster convergence and efficient hyperparameter transfer. Moreover, large-scale experiments with PixArt-α and MMDiT reveal that µP enables superior performance with minimal tuning cost, establishing µP as a robust and efficient framework for scaling diffusion Transformers.

**Questions:**

Q1: Can the authors better justify the novelty of their work, given that the µP formulation in diffusion Transformers (e.g., DiT) appears structurally similar to vanilla Transformers with only minor conditioning changes (e.g., class + timestep via AdaLN)? What specific technical challenges or non-trivial insights distinguish this extension?

Q2: Are the performance gains still significant when classifier-free guidance (CFG) is applied during generation? Please provide comparisons with and without CFG, especially for results in Figure 4 and Table 3.

Q3: In Figure 5, what is the rationale for including MMDiT in the REPA comparison, and how is it related? Can the authors clarify the meaning and role of "base warm-up interactions" and ensure that all such terms are well-defined in the main text?

Q4: The "base learning rate" subplot in Figure 5 appears noisy and lacks a clear justification for selecting 2.5e-4 as optimal. Can the authors provide more systematic tuning experiments or ablation studies to support this choice?

Q5: While Figure 6 shows lower training loss, it is unclear how this correlates with generation quality. Can the authors provide additional evaluation on downstream generation quality (e.g., FID, CLIP score, human studies) to confirm whether improved training loss leads to better outputs during inference?

**Ethical Concerns:**

["NO or VERY MINOR ethics concerns only"]

**Final Justification:**

After reviewing it along with the other reviews, I find that the authors have addressed some of my concerns, particularly regarding experimental clarity and their justification for µP in diffusion Transformers.

However, I still believe the novelty remains limited, as also pointed out by Reviewer sRBx. While the theoretical extension and large-scale empirical study are appreciated, the overall contribution feels incremental.

I adjusted my score from 2 to 3 to reflect the partial resolution of concerns, but I remain cautious about the paper’s overall significance.

**Limitations:**

Yes

**Quality:**

2

**Strengths And Weaknesses:**

**Strengths**

+ The paper explores the adaptation of Maximal Update Parametrization (µP) to diffusion Transformers, which could be beneficial for scaling vision generative models with lower hyperparameter tuning costs.

+ Some experimental results, particularly on DiT, validate the applicability of µP in the vision domain, indicating a promising direction for efficient large-scale training.

**Weaknesses**

+ The novelty of the work is questionable. While the authors initially suggest that µP’s hyperparameter transferability may not hold for diffusion Transformers, they ultimately prove that the µP formulation aligns with that of vanilla Transformers. This result is not particularly surprising, given the structural similarities—DiT mainly comprises standard Transformer blocks with adaptive layer normalization conditioned on class and timestep embeddings. As such, applying µP here seems like a straightforward extension rather than a novel contribution.

+ The reported performance gains appear marginal. For instance, in Figure 4, the improvements seem modest and are likely measured without classifier-free guidance (CFG). With CFG applied, the performance gap might diminish further. Similar trends are observed in Table 3 across several evaluation metrics.

+ In Figure 5, the inclusion of MMDiT in relation to the REPA method seems unclear and insufficiently justified. Additionally, terms like “base warm-up interactions” are introduced without adequate explanation, making it difficult to understand their relevance. The “base learning rate” subplot also appears noisy and lacks clear justification for the chosen optimal value (e.g., 2.5e-4).

+ Figure 6 shows lower training loss, but this does not necessarily translate to better generative performance during inference. More compelling qualitative or quantitative evidence is needed to support the claim of improved generation.

---

> ### Author Rebuttal · Authors · 2025-07-31
>
> We thank Reviewer mn36 for the feedback and valuable comments.
>
> ## Weakness 1 & Question 1: novelty
>
> Thank you for the question.  We respectfully offer the following clarifications.
>
> From a theoretical perspective, **it is not obvious that the µP form remains valid when the network architecture is modified**. In fact, recent evidence shows that architectural changes can break µP’s conditions (e.g., the Mamba architecture [54]). Therefore, a new rigorous derivation for diffusion Transformers is necessary to ensure that additional modules do not invalidate the theoretical guarantees. **As acknowledged by Reviewers hm23, "this provides a principled basis for the experimental work, moving beyond simple heuristics"**.
>
> Beyond the theoretical contribution, **our large-scale empirical study is also significant and non-trivial**:
>
> **Significance**. Training large-scale diffusion Transformers requires enormous compute resources (e.g., ~100M dollars for Sora), while still lacking a principled way to choose hyperparameters at scale. This bottleneck limits the development of large vision foundation models. **Our work gives a preliminary solution to this significant problem** by showing µP as a principled method to obtain good hyperparameters with small tuning cost (e.g., 3% for MMDiT-18B).
>
> **Non-trivial**. Prior to our work, µP was rarely applied in large-scale diffusion model pretraining. This was mainly because there was **no rigorous µP theory** for diffusion Transformers, **nor clear empirical evidence** that µP would ensure stable and effective hyperparameter transfer for these models. Based on our rigorous µP theory, we conducted large-scale experiments, which demonstrate that µP indeed enables reliable and effective hyperparameter transfer for diffusion Transformers, **providing the vision community a principled approach for scaling diffusion Transformers efficiently**.
>
> We believe this bridge from theoretical consistency to robust practical evidence distinguishes our work from prior µP studies and provides meaningful value to the vision and generative modeling communities.
>
>
> ## Weakness 2 & Question 2: comparisons with and without CFG
>
> Thank you for the question. We respectfully offer the following clarifications.
>
> ### Performance gains appear marginal
>
> We would like to emphasize that our main goal is not to improve performance significantly, but to **provide a principled, low-cost approach to find hyperparameters** that match or even surpass human-tuned baselines—**especially in large-scale training settings** (e.g., only 3% FLOPs for MMDiT-18B).
>
> ### Results with and without CFG
>
> To clarify, **for each experiment, we followed the exact setup of the corresponding baseline paper to ensure fair comparison**. Specifically, DiT does not use CFG during training evaluations (see Table 4 in [48]), so our DiT hyperparameter search in Figure 3 also does not include CFG. In contrast, PixArt-α uses CFG during evaluation (see Table 2 in [8]), so our hyperparameter search for PixArt-α (Table 2) includes CFG as well.
>
> Because CFG is an important factor during inference and **affects the optimal hyperparameters**, reproducing results in Figure 4 with CFG for DiT would principly require re-running the full proxy search and retraining the DiT-XL-2 under the new searched hyperparameters, which would cost roughly **300 A100 GPU days** (as detailed in Appendix E.2.4). Unfortunately, this is not feasible within the rebuttal period, but we are willing to include these additional runs in the final version.
>
> Nevertheless, **our current results already cover both settings: µP works well without CFG (DiT) and with CFG (PixArt-α, MMDiT)**. In both cases, we observe stable hyperparameter transferability and performance improvements over the baseline.
>
> ## Weakness 3 & Question 3: unclear REPA loss and warm-up
>
> Thank you for raising this important point. Though we mention these hyperparameters in Section 5.2.1 (lines 260–264), we agree they could be explained in more detail. Below, we provide a clearer clarification:
>
> 1. The REPA loss [69] is an auxiliary regularization term widely used in diffusion Transformer to accelerate training, which explicitly aligns the diffusion model representation with powerful pretrained visual representation (e.g., DINOv2). Therefore, we include REPA loss and aim to **provide practitioners a suitable REPA weight for large-scale diffusion Transformers training**.
> 2. In fact, **there is no term “base warm-up interactions” in our paper**. The term used is “base warm-up iteration”, which refers to the warm-up steps used in the learning rate schedule. This is standard practice for large models, where the learning rate starts small and gradually grows up to its target value to improve stability at the start of training.
>
> Thank you for highlighting this. We will revise the text to define REPA weight and warm-up iteration more explicitly in the main paper so that their meaning and role are unambiguous.
>
> ## Weakness 3 & Question 4: selection of learning rate
>
> Thank you for raising this practical question.
>
> 1. **We follow the established practice in the µP literature** [12, 27, 66], where the base hyperparameter is typically selected by identifying the value that aligns with the lowest envelope of the training loss plot. In our case, **we observed that the envelope near 2.5e-4 was stable and close to optimal**, so we chose 2.5e-4 as a representative value within this range.
> 2. As noted in lines 274–275 and detailed in Appendix Table 15, **we did perform a more systematic check**: we ran additional training with five candidate learning rates around 2.5e-4, each for 100K steps (the original sweep in Figure 5 covered 30K steps). The results confirmed that **learning rates near 2.5e-4 (2e-4~4e-4) all produced robust and strong performance**, with 2.5e-4 yielding the best overall results.
>
> We will clarify the envelope selection logic and highlight the extended ablation in the final version.
>
> ## Weakness 4 & Question 5:  evaluation on downstream generation quality
>
> Thank you for raising this valid point about generation quality.
>
> To address this, **we have already included human evaluation results in the submission**. As described in **Section 5.2.1 (lines 280–287) and Table 4**, we conducted a comprehensive human study where 10 human experts compared outputs from µP and baseline models. The µP-trained MMDiT model outperformed the baseline in human preference scores, confirming that the observed training loss reduction translates to better sample quality. **The visual comparison is also presented in Figure 10 in Appendix E.4.6**.
>
> In addition, **we now provide the Geneval score, which is a standard quantitative metric used for text-to-image models** (e.g., Stable Diffusion 3 [15]). The **overall** Geneval score for the µP-trained MMDiT model is higher than the baseline, further supporting the claim that lower training loss yields better generative performance at inference.
>
>
> | Method       | Overall $\uparrow$ | Single | Two | Counting | Colors | Position | Color Attribution |
> | ------------ | -------- | -------- | -|-|-|-|-|
> | MMDiT-18B     | 0.8154     | 99.38    | 93.69 | 81.88 | 88.03 | 57.50 | 68.75|
> | MMDiT-μP-18B     | **0.8218**     | 99.38    | 94.44 | 79.69 | 88.83 | 62.25| 68.50|
>
> Thank you for suggesting this important clarification. We agree that these links can be presented more clearly and will highlight them alongside Figure 6 in the final version.

---

> > ### Comment · Reviewer_mn36 · 2025-08-05
> >
> > Thanks for sharing the rebuttal. After reviewing it along with the other reviews, I find that the authors have addressed some of my concerns, particularly regarding experimental clarity and their justification for µP in diffusion Transformers.
> >
> > However, I still believe the novelty remains limited, as also pointed out by Reviewer sRBx. While the theoretical extension and large-scale empirical study are appreciated, the overall contribution feels incremental.
> >
> > I will adjust my score from 2 to 3 (borderline RJ) to reflect the partial resolution of concerns, but I remain cautious about the paper’s overall significance.

---

> > > ### Author Response · Authors · 2025-08-06
> > >
> > > Thank you for your positive feedback! We’re glad that our response helped address some of your concerns, and we appreciate your consideration in raising the score. We’d like further to clarify the significance and contribution of our work:
> > > 1. Significance: Training large-scale diffusion Transformers requires enormous compute, yet the field still lacks a principled and scalable method for hyperparameter selection. This bottleneck has become a key challenge in advancing large vision foundation models. Our work gives a preliminary solution to this significant problem.
> > > 2. Contribution: We present the rigorous proof of the µP form for diffusion Transformers, establishing a theoretical foundation beyond heuristic usage. Based on this, we conduct large-scale experiments across architectures and data regimes, demonstrating that µP enables stable and effective hyperparameter transfer in practice.
> > >
> > > In summary, our paper offers the vision community a principled approach for reliably scaling diffusion Transformers.

---

### Official Review · Reviewer_sRBx · 2025-07-01

**Clarity:** 3
**Significance:** 3
**Originality:** 2
**Rating:** 4
**Confidence:** 4

**Summary:**

This paper generalizes the standard Maximal Update Parametrization, short for μP, to diffusion
Transformers and validate their effectiveness through large-scale experiments.
They rigorously prove that μP of mainstream diffusion Transformers, including DiT, U-ViT, PixArt-α, and MMDiT, aligns with that of the vanilla Transformer, enabling the direct application of existing μP methodologies. The article systematically demonstrates that DiT-μP enjoys robust HP transferability. Furthermore, they validate the effectiveness of μP on text-to-image generation by scaling PixArt-α from 0.04B to 0.61B and MMDiT from 0.18B to 18B. In both cases, models under μP outperform their respective baselines while requiring small tuning cost.

**Questions:**

1. This investigation of the Maximal Update Parametrization method for the diffusion transformer is valuable.
However, I believe the author should focus on distinguishing between vanilla Transformers and diffusion Transformers and develop specific characteristics for the Maximal Update Parametrization method that address this distinction. Otherwise, the author could consider providing a unified theoretical formulation for Maximal Update Parametrization that encompasses both vanilla Transformers and diffusion Transformers.

2. I think you can verify the convergence speed for Maximal Update Parametrization method on PixArt-α and MMDiT models.

3. I think you can scale the model from a small number of parameters to a large number of parameters in Dit models and test its effectiveness.

4. If the experiment scales the PixArt-α-μP starting from 0.02B or 0.08B parameters, how would the results change? If it scales the MMDiT-μP starting from 0.06B or 0.5B parameters, what differences might occur?

**Ethical Concerns:**

["NO or VERY MINOR ethics concerns only"]

**Final Justification:**

The author's response to the reviewers has addressed many of my concerns. I believe this work can help the research community explore the generative model more efficiently. However, the article still faces some unresolved issues. I will list them above.

# solved issues (These responses have considered the rebuttals with reviewers hm23 and AKwD)
1.  The method drastically reduces the cost of finding optimal hyperparameters for billion-parameter models.

2.  By transferring HPs from small proxy models, the tuning cost for the 18B MMDiT model was reduced to just 3% of the cost of traditional manual tuning by experts. Meanwhile, the method achieved 2.9x faster convergence for DiT-XL-2 and outperformed baselines in large-scale text-to-image tasks. These results make large-scale model development more accessible and efficient.

3. The approach is validated across multiple modern architectures (DiT, PixArt-α, MMDiT), different scaling dimensions (model width, batch size, number of training steps), and various generative tasks, demonstrating its robustness and wide applicability.

The experiments show clear, substantial improvements.

# unsolved issues (These responses have considered the rebuttals with reviewers sRBx and mn36)
1. The original contribution of the work as a research article could be more substantial. The article proves the µP form in the new architecture without more theoretical insight. This seems like a straightforward extension rather than a novel contribution.

Although a solid theoretical proof is essential, I believe that beneficial empirical results are more helpful for the community.

 Based on the above comments, I am leaning towards increasing my score to 4.

**Limitations:**

yes

**Quality:**

2

**Strengths And Weaknesses:**

# Strengths

The article presents systematic experiments on the effectiveness of the Maximal Update Parametrization method for diffusion transformers. Including DiT-XL-2-μP with transferred learning rate achieves 2.9× faster convergence than the original DiT-XL-2. And only 5.5% of one training run for PixArt-α, 3% of consumption by human experts for MMDiT-18B outperformed their respective baselines. Moreover, the article is well written and organized.


# Weaknesses

The original contributions of the article are not sufficient. The article reads like a technical report rather than a research paper. The author just verified the Maximal Update Parametrization method on the diffusion Transformer models. Although they propose a new theorem to ascertain the effectiveness of Maximal Update Parametrization on the diffusion transformer, it only demonstrates that the diffusion transformer can be represented within the NE⊗OR⊤ Program. The new theorem does not offer any new theoretical insight. Furthermore, the author stated that the DIT includes some modules different from the standard transformer, such as Scale, Shift, T-emb, and y-emb, as highlighted in Figure 2. However, these modules appear to have no impact on the proof of DIT within the NE⊗OR⊤ Program, which is shown in the appendix.
Based on the above content, the article lacks enough contribution and novelty.

---

> ### Author Rebuttal · Authors · 2025-07-31
>
> We thank Reviewer sRBx for the feedback and valuable comments.
>
> ## Weakness: novelty
>
> Thank you for raising these concerns about the contribution and novelty of our work. We respectfully offer the following clarifications.
> ### Significance and contribution
>
> First, we would like to emphasize that **verifying a theory or method does not imply insufficient contribution**, especially when the setting is practically important and the conclusions can impact a broad audience. For example, ViT [a] successfully changed how the community builds vision models by systematically verifying that the Transformer architecture could be transferred to vision models. In this way, **introducing and verifying µP in the large-scale diffusion Transformer is also significant and non-trivial**.
>
> **Significance**. Training large-scale diffusion Transformers requires enormous compute resources (e.g., ~100M dollars for Sora), while still lacking a principled way to choose hyperparameters at scale. This bottleneck limits the development of large vision foundation models. **Our work gives a preliminary solution to this significant problem** by showing µP as a principled method to obtain good hyperparameters with small tuning cost (e.g., 3% for MMDiT-18B).
>
> **Non-trivial**. Prior to our work, µP was rarely applied in large-scale diffusion model pretraining. This was mainly because there was **no rigorous µP theory** for diffusion Transformers, **nor clear empirical evidence** that µP would ensure stable and effective hyperparameter transfer for these models. **We take a first step to address this gap, both theoretically and empirically**. First, we formally proved the µP form for diffusion Transformers. **As acknowledged by Reviewers hm23, "this provides a principled basis for the experimental work, moving beyond simple heuristics"**. We then conducted large-scale experiments, which demonstrate that µP indeed enables reliable and effective hyperparameter transfer for diffusion Transformers, **providing the vision community a principled approach for scaling diffusion Transformers efficiently**.
>
> ### Proof of DiT
> On the role of additional modules: we do not fully agree that modules like Scale, Shift, T-emb, and y-emb “have no impact” on the proof. **As detailed in Appendix C (e.g., lines 664–721)**, we provide **explicit derivations** showing how these components are mapped and handled within the NE⊗OR⊤ Program. These **module-specific derivations** are essential to ensure that the entire DiT architecture adheres to the same µP consistency.
>
> In summary, we believe our paper contributes both practically and theoretically: it rigorously extends µP theory to an important new domain, and provides the community with a validated, scalable methodology that is easy to adopt. We sincerely thank the reviewer for pushing us to better clarify these contributions and will emphasize them more clearly in the final version.
>
> [a] An Image is Worth 16x16 Words: Transformers for Image Recognition at Scale, ICLR, 2021
>
> ## Question 1: unified theoretical formulation
>
> Thank you very much for this constructive suggestion. In fact, our theoretical proof is built precisely on the latter idea: we **unify the µP of diffusion Transformers and the classic µP (e.g., vanilla Transformers)** by leveraging the NE⊗OR⊤ Program framework [65] to represent the forward pass of diffusion Transformers.
>
> We will emphasize this point more clearly in the final version to highlight that our theorem indeed offers the requested unification. Thank you for pointing this out and helping us clarify this aspect of our contribution.
>
> ## Question 2: convergence speed on PixArt-α and MMDiT
>
> Thank you for this suggestion. In fact, **we have already included empirical evidence for the faster convergence speed of µP on PixArt-α and MMDiT in the current version**. Table 3 and Figure 6 show the performance trajectories during training, clearly indicating that µP consistently achieves better performance than the baselines throughout the training process. This directly demonstrates that µP enables faster convergence for these models as well.
>
> We would also like to emphasize that our main goal is not merely to maximize convergence speed, **but to provide a principled, low-cost approach to find hyperparameters** that match or even surpass human-tuned baselines—**especially in large-scale training settings** (e.g., only 3% FLOPs for MMDiT-18B).
>
> We appreciate the reviewer’s point that this could be made clearer. We will improve the visualization and discussion in the final version to more explicitly highlight the convergence speed improvements enabled by µP.
>
> ## Question 3: scaling from small DiT model to large DiT model
>
> Thank you for this suggestion. As an initial exploration of µP for Diffusion Transformers, we followed the classical µP theory [63, 66] by scaling width while keeping depth fixed. **We did not scale from a small DiT variant (e.g., DiT-S) to a large DiT variant (e.g., DiT-XL) because their depths differ significantly**, which does not satisfy the assumptions required for directly verifying our theoretical guarantee.
>
> We note that the recent work [b] proposes ideas to enable hyperparameter transfer across depth in the context of LLMs, which is an interesting direction for future work on scaling the depth of diffusion Transformers.
>
> We thank the reviewer for highlighting this point and will mention this potential extension more clearly in the final version.
>
> [b] Don't be lazy: CompleteP enables compute-efficient deep transformers, 05/2025
>
> ## Question 4: scaling PixArt-α/MMDiT from different scale
>
> Thank you for this practical suggestion. We run additional experiments for PixArt-α-µP here. **As a result, we found that the base learning rate searched on the 0.04B proxy model can be transferred to a larger model (e.g., 0.087B)**.  Concretely, we keep the same setup as in Table 2 in the submission but vary the number of attention heads to get different parameter scales. The results are:
>
> | head| width | Params |$\log_2$(lr) | Training loss $\downarrow$ | Geneval $\uparrow$ |
> |-| -|-|-------- | -------- | -------- |
> |2   | 144 |   0.01B  |  -9  | NaN     |  NaN    |
> |2   | 144 |   0.01B  |  -10  |   0.1539   |  0.03   |
> |**2**   | **144** |   **0.01B**  | **-11**  |   **0.1537**   |  **0.046**    |
> |2   | 144 |   0.01B  |  -12  |   0.155   |  0.038    |
> |4   | 288 |   0.04B  |  -9  | NaN     |  NaN    |
> |**4**   | **288** |   **0.04B**  |  **-10**  |   **0.1493**   | **0.083**    |
> |4   | 288 |   0.04B  |  -11  |   0.1494   |  0.078    |
> |4   | 288 |   0.04B  |  -11  |   0.1494   |  0.078    |
> |6   | 432 |   0.087B  |  -9  | NaN     |  NaN    |
> |**6** | **432**   |   **0.087B**  |  **-10**  |   **0.1475**   |  **0.134**    |
> |6     | 432  |   0.087B  |  -11  |   0.1476   |  0.132    |
> |6     | 432  |   0.087B  |  -12  |   0.1487   |  0.062    |
>
> As shown above, the optimal base learning rate for 0.087B matches the 0.04B proxy ($2^{-10}$), **confirming that our base learning rate in the main paper transfer well to wider models**. For the 0.01B proxy, the best learning rate shifts **slightly** ($2^{-11}$), which is expected because µP theory predicts that base hyperparameter transfer is stable when the width is sufficiently large (our experiments in the main paper suggest ~256–512 width works well).
>
> This new test supports our claim that µP enables robust hyperparameter transfer. Due to computational limits, we could not repeat this for MMDiT within the rebuttal phase, but we expect similar trends to hold.
>
> We appreciate this suggestion and will include a clearer discussion of the practical lower-bound proxy scale in the final version.

---

> > ### Comment · Reviewer_sRBx · 2025-08-03
> > **Response To Author**
> >
> > Thank you for your response. Your response to the reviewers has addressed many of my concerns. Although I think the original contribution of your work as a research article could be more substantial, I find your work insightful, the empirical results are sufficient, and they can adequately support your claims. I am leaning towards increasing my score to 4.

---

> > > ### Author Response · Authors · 2025-08-03
> > > **Thank you**
> > >
> > > Thank you for your positive feedback! We are delighted that our response resolved your concerns and appreciate your consideration in raising the score. We will definitely add the new discussions and results in the rebuttal to the final version.

---

### Official Review · Reviewer_hm23 · 2025-07-01

**Clarity:** 3
**Significance:** 3
**Originality:** 3
**Rating:** 5
**Confidence:** 4

**Summary:**

This paper addresses a critical bottleneck in developing large-scale generative models: the prohibitive cost of hyperparameter (HP) tuning. The authors propose applying Maximal Update Parametrization (µP), a technique successful in Large Language Models (LLMs), to Diffusion Transformers.
The work makes three core contributions. First, it provides a rigorous theoretical proof that mainstream Diffusion Transformer architectures (including DiT, U-ViT, PixArt-α, and MMDIT) share the same µP formulation as standard vanilla Transformers. This key finding allows the direct application of existing µP methodologies. Second, the paper systematically validates that under µP, optimal hyperparameters (like the learning rate) are stable and can be reliably transferred from small "proxy" models to large-scale models.
Finally, the authors demonstrate significant practical benefits. A DiT-XL-2 model trained with a transferred learning rate achieves 2.9x faster convergence. Furthermore, when scaling up text-to-image models like PixArt-α and MMDiT (up to 18B parameters), the µP approach consistently outperforms baselines while requiring only a fraction of the typical tuning cost—as little as 3% of the effort of manual expert tuning.

**Questions:**

- Your theoretical proof confirms that Diffusion Transformers share the same µP formulation as vanilla Transformers. However, the denoising objective in diffusion models is fundamentally different from the autoregressive, next-token prediction objective in LLMs where µP was popularized. Did you observe any instances where this difference in training objective affected HP transfer stability, particularly when transferring across a large number of training steps?

- The paper scales model width by increasing the number of attention heads while keeping the head dimension fixed. Does the µP framework offer any theoretical guidance on this choice versus the alternative of scaling the head dimension itself? Have you empirically explored if one scaling strategy leads to more stable HP transfer than the other?

- You note an interesting difference in optimal learning rates: for DiT/PixArt-α (trained for multiple epochs), the optimal LR was near the stability limit, while for the 18B MMDIT (trained for a single pass), it was not. Could you elaborate on why this occurs?

**Ethical Concerns:**

["NO or VERY MINOR ethics concerns only"]

**Final Justification:**

While I understand that the theoretical contribution could be more substantial, I tend to accept this paper since i) it is rigorous and consistent from theoretical groundings to empirical experiments; ii) It offers practical value by helping to reduce the costly trial-and-error process involved in training large-scale diffusion models; iii) the experiments are comprehensive to support the idea and also serve as a user guide for practitioners.

**Limitations:**

yes

**Quality:**

3

**Strengths And Weaknesses:**

Strengths:
- The paper is built on a rigorous proof (Theorem 3.1) that formally extends µP theory to cover several state-of-the-art Diffusion Transformer architectures. This provides a principled basis for the experimental work, moving beyond simple heuristics.

- The method drastically reduces the cost of finding optimal hyperparameters for billion-parameter models. By transferring HPs from small proxy models, the tuning cost for the 18B MMDiT model was reduced to just 3% of the cost of traditional manual tuning by experts, making large-scale model development more accessible and efficient. The experiments show clear, substantial improvements. Achieving 2.9x faster convergence for DiT-XL-2  and outperforming baselines in large-scale text-to-image tasks  are powerful validations of the method's effectiveness.

- The approach is validated across multiple modern architectures (DiT, PixArt-α, MMDiT), different scaling dimensions (model width, batch size, number of training steps), and various generative tasks, demonstrating its robustness and wide applicability.

Weaknesses:
- The formal proof in the appendix relies on an assumption (Assumption C.1) that simplifies model architectures, for instance by treating certain fixed dimensions as 1 and assuming all layers share the same width. While the paper notes this is standard for such proofs, it is a deviation from the true complexity of the models being analyzed.
- The paper successfully uses small "proxy" models for tuning, but as acknowledged in the conclusion, it does not investigate the optimal design of these proxy tasks. There is no analysis of the trade-off between the size (and cost) of the proxy model and the quality of the transferred HPs, leaving this as an open question for practitioners.
- The experiments primarily focus on transferring the learning rate. While the learning rate is often the most critical HP, the paper does not extensively explore the transferability of other important parameters like weight decay, dropout, or optimizer-specific settings (e.g., Adam's betas), which can also significantly impact performance.

---

> ### Author Rebuttal · Authors · 2025-07-30
>
> We thank Reviewer hm23 for the positive support and valuable comments, which can definitely improve the quality of this paper.
>
> ## Weakness 1: assumption in proof
>
> Thank you for raising this thoughtful point. **In principle, the proof technique naturally extends to any finite fixed dimension (>1) and to hidden layers with different widths (non-uniform infinite-width setting)**.
>
> To show that our theory can be extended to any finite fixed dimensions, we take the ouput projection layer $\boldsymbol{y} = \boldsymbol{W} \boldsymbol{x}$ as an example (corresponding to line 717-718 at Appendix C.1), where the output dimension is any finite $d$, $\boldsymbol{W} \in \mathbb{R}^{d\times n}$, and $\boldsymbol{x} \in \mathbb{R}^{n}$. To represent the operation in the NE⊗OR⊤ Program, we denote the $i$-th row of $\boldsymbol{W}$ by $\boldsymbol{w}^{(i)} \in \mathbb{R}^n$, then the output projection layer can be written as
> \begin{align*}
>  \widetilde{c}^{(i)} _\alpha = \psi _1([\boldsymbol{w}^{(i)}, \boldsymbol{x}] _{\alpha:}) = n {w}^{(i)} _\alpha x _\alpha, (\text{OuterNonlin})  \quad \quad
>  {y} _{i} = \frac{1}{n} \sum _{\alpha=1}^n \widetilde{c}^{(i)} _\alpha.  (\text{Avg})
> \end{align*}
>
> For the non-uniform infinite-width setting, the Avg, MatMul, and OuterNonlin operators in Appendix B.2 simply need to be extended to accept input tensors with any $\Theta(n)$ dimension rather than a single shared width $n$.
>
> Furthermore, **please see Section 2.9 in TP-4b [65], which provides a more detailed discussion of both these generalizations** and shows how the tensor program framework naturally accommodates them.
>
> We will clarify this point in the appendix to make explicit how these assumptions can be relaxed to better reflect the practical model architecture. We appreciate the reviewer for pointing this out and helping us improve the presentation.
>
> ## Weakness 2: optimal design of proxy tasks
>
> We thank the reviewer for this nice suggestion. We note that determining the optimal design of proxy tasks is a rarely explored question in the µP literature [66, 12, 27, 40, 22, 39, 71]. In this paper, **our empirical results on three different diffusion Transformers and datasets provide a practical starting point for the vision community**:
> 1. A proxy model width of approximately 256–512 works reliably for stable hyperparameter transfer.
> 2. The data scale for proxy tuning typically needs to reach about 1/10 to 1/6 of the final pretraining dataset to yield robust hyperparameter transfer.
>
> In addition, to guide future work for determining the optimal design of proxy tasks, we **suggest a concrete experiment design**:
> 1. Select a target large-scale diffusion Transformer (e.g., MMDiT-18B with a width of 5120).
> 2. Construct multiple proxy tasks varying in width from 64 up to 5120, and data from 10% up to 100% of the target dataset.
> 3. For each proxy task, tune the base HPs and evaluate their transferability to the target task.
> 4. Analyze the relationship between the proxy task scale and final performance to identify the optimal proxy scale.
>
> Given our current computational budget, we could not conduct this sweep, but we strongly encourage future work to investigate this direction. We will add this discussion to the final version to better highlight this practical research frontier. Thank you for encouraging us to clarify this.
>
> ## Weakness 3: transferability of other HP
>
> Thank you for this thoughtful comment. We agree that investigating the transferability of other hyperparameters is important. **In Figure 7 at Appendix E.2.2, we also empirically verified the transferability of the output multiplier**, which is an important hyperparameter in µP theory and practice [66, 40, 5]. We will make this more visible in the main text to avoid confusion.
>
> In addition, although we did not run an explicit transferability study for each individual base hyperparameter in MMDiT, it is worth noting that the hyperparameters found on the proxy model transferred successfully to the target 100× larger model, outperforming hand-tuned settings. **This strongly suggests that other base hyperparameters are reasonably stable under µP as well**.
>
> We are open to the reviewer’s further feedback. Thank you again for raising this practical point and helping us clarify our contribution!
>
> ## Question 1: influence of denoising objective
>
> Thank you for raising this thoughtful question. Testing whether the different training objective and data domain affect the hyperparameter transferability of µP is also one of the key motivations behind our work. In fact, across all three of our experimental settings, **we did not observe any instability in base HP transfer caused by the denoising objective**, even when transferring across a large number of training steps (from 200k steps to 2400k steps in DiT, and from 30k steps to 200k steps in MMDiT).
>
> We will clarify this empirical observation more explicitly in the final version. We appreciate the reviewer for highlighting this point.
>
> ## Question 2: why scaling number of attention heads
>
> Thank you for raising this insightful question. **Our choice to scale model width by increasing the number of attention heads while fixing the head dimension follows both theoretical and practical guidance**.
>
> 1. Theoretically, recent theoretical advance [5] shows that both scaling strategies are consistent with the transferability condition required by µP. However, they also reveal an important difference: **when the head dimension tends to infinity, multi-head self-attention can collapse to single-head self-attention, losing the diversity of attention patterns**. In contrast, scaling the number of heads avoids this degeneracy and preserves the expressivity of the multi-head mechanism.
> 2. Empirically, the TP-V paper [66] (please see Figure 4 and Figure 13) shows that both scaling strategies yield stable base HP transfer under µP. However, as we mentioned in line 120, **well-known models in real-world large-scale practice like Llama3 [21] and MiniCPM [27] favor increasing the number of heads** rather than the head dimension.
>
> Given these theoretical and empirical insights, we chose this widely supported scaling strategy in our experiments. A systematic empirical comparison of the two strategies under µP for diffusion Transformers is an interesting direction for future work, and we will clarify this point in the discussion section in the final version.
>
> ## Question 3: optimal learning rate and stability edge
>
> Thank you for highlighting this intriguing point. This phenomenon has also been noted in recent studies [58] and remains an interesting open question in large-scale training.
> 1. For multiple-epoch training, it is well established that the optimal learning rate often lies near the stability limit [a; b; c]. Larger learning rates is thought to introduce beneficial gradient noise, which can help guide optimization towards flatter minima that generalize better. Several theoretical perspectives attempt to explain this, including the flat minima hypothesis [d] and the edge of stability framework [f].
> 2. For single-pass training, recent work [58] in the context of LLM pretraining has also pointed out that the optimal learning rate tends to be away from the stability limit. This suggests a fundamental difference in optimization dynamics between single-pass and multi-epoch regimes, although the precise reason remains an open question. Intuitively, since the gradient signal for any individual sample is not revisited, the training must be more conservative to maintain stability.
>
> Overall, this is a meaningful difference that highlights the complex interaction between optimal learning rate, model scale, and training strategy. We agree it is a valuable topic for future theoretical investigation, and we will add a brief discussion of this point in the final version.
>
> [a] Towards explaining the regularization effect of initial large learning rate in training neural networks, NeurIPS, 2019
>
> [b] The large learning rate phase of deep learning: the catapult mechanism, 2021
>
> [c] Finite versus infinite neural networks: an empirical study, NeurIPS, 2020
>
> [d] On the Relation Between the Sharpest Directions of DNN Loss and the SGD Step Length, 2018
>
> [f] Gradient Descent on Neural Networks Typically Occurs at the Edge of Stability, ICLR, 2021

---

> > ### Comment · Reviewer_hm23 · 2025-08-05
> >
> > My concerns have been addressed by the authors' rebuttal. And I would like to keep my rating.

---

> > > ### Author Response · Authors · 2025-08-05
> > > **Thank you**
> > >
> > > Thank you for your positive feedback! We are glad that our response addressed your concerns. We will definitely add the new discussions in the rebuttal to the final version.

---

### Official Review · Reviewer_AKwD · 2025-07-02

**Clarity:** 4
**Significance:** 3
**Originality:** 3
**Rating:** 5
**Confidence:** 2

**Summary:**

The paper extends $\mu P$ theory to the mainstream diffusion transformer family. The authors rigorously prove the scaling compatibility with vanilla transformers, and derives an implementation recipe for robust HP transfer. Extensive experiments on class-conditional model DiT and text-conditional models on PixArt-$\alpha$ and MMDiT show faster convergence, lower tuning cost and better FID/GenEval than carefully tuned baselines.

**Questions:**

See "weakness" section.
I am willing to change my score if my concerns are addressed.

**Ethical Concerns:**

["NO or VERY MINOR ethics concerns only"]

**Final Justification:**

I appreciate the authors' rebuttal and have examined the feedback from the other reviewers together with the newly provided experiments. I am inclined to recommend acceptance for this paper: 1. Although the paper is not the first academic attempt to apply $\mu P$ to diffusion transformers operating on image data, the breadth and depth of the empirical investigation offer comprehensive insights that will benefit the broader community. 2. Although the theoretical framework represents an extension of prior works on vanilla transformer, the paper establishes a reasonable degree of theoretical rigor that supports the methodological contributions.

Taking these points together, my final score falls between “borderline accept” and “accept,” leaning toward “accept,” provided that the authors incorporate the promised clarifications (including the discussion of AuraFlow and the supplementary experiments provided in response to reviewers sRBX and hm23).

**Limitations:**

Yes

**Quality:**

3

**Strengths And Weaknesses:**

Strengths:
1. The paper shows solid theoretical proof of diffusion transformers fitting the $NE \otimes OR^{\top}$ program, and distinguishes it from prior uses of $\mu P$ in diffusion transformers without a theoretical guarantee.
2. This work also spans various image generation models and evaluates the performance with standard metrics plus human expert checks.
3. The writing is clear and easy to follow.

Weaknesses:
1. The paper fails to cite and discuss the AuraFlow v0.1 blog (cloneofsimo, Team Fal, July 2024). They have applied $\mu P$ empirically to an MMDiT-style model for zero-shot LR transfer.[1]
2. The warmup-straedy-decay learning rate schedule is a widely used schedule for the training of diffusion transformers. It is unclear whether $\mu P$ interacts benignly with schedules like these schedules. It will be beneficial if the authors could at least provide intuitive thoughts to this.

[1] Introducing AuraFlow v0.1, an Open Exploration of Large Rectified Flow Models. cloneofsimo, Team fal. https://blog.fal.ai/auraflow/

---

> ### Author Rebuttal · Authors · 2025-07-30
>
> We thank Reviewer AKwD for the positive score and valuable comments, which can definitely improve the quality of this paper.
>
> ## Weakness 1: missing related blog
>
> Thank you very much for pointing this out. We will certainly add a citation and discussion of the AuraFlow v0.1 blog in the final version. We acknowledge that while we carefully surveyed peer-reviewed publications that use µP for diffusion models [22,52], we unfortunately overlooked the insightful AuraFlow v0.1 blog.
>
> Compared to AuraFlow v0.1, our work makes additional non-trivial contributions:
> 1. We provide a rigorous theoretical proof for mainstream diffusion Transformers, thereby justifying the validity of µP for this family of models.
> 2. We systematically validate the hyperparameter (HP) transferability of diffusion Transformers under µP, across multiple widths, batch sizes, and training steps.
> 3. For MMDiT in particular, we conduct a more extensive search over multiple HPs and scale the model up to 18B parameters, providing detailed intermediate results and comparisons with human-tuned baselines. We believe these results offer principled and reproducible insights for future large-scale scaling of diffusion Transformers.
>
> We will explicitly include a discussion of AuraFlow v0.1 in the related works section of the final version. We thank the reviewer again for highlighting this relevant prior effort!
>
> ## Weakness 2: other learning schedules
>
> We thank the reviewer for raising this insightful question.
>
> Yes. According to our theoretical result (Theorem 3.1), which rigorously identifies the µP form of diffusion Transformers, and based on existing µP theory (please see Table 2 and Figure 4 in TP-V [66]), optimizer-related schedules and HPs, including the warmup steps, schedules (e.g., constant, cosine, and warmup-steady-decay), and maximum learning rate, can in principle be treated as HPs that can be searched in the proxy model and transferred to the large model. We will try to present a new corollary for different learning schedules in the final version.
>
> Due to limited computational resources, we have not yet conducted an extensive search over schedules in this work, but we see this as a promising and scientifically meaningful direction. Our study provides a starting point for future works to investigate the optimal schedules for large-scale diffusion Transformers.
>
> We will clarify this point and add a discussion on this possible extension in the discussion section in the final version. Thank you again for pointing it out!

---

> > ### Comment · Reviewer_AKwD · 2025-08-07
> >
> > I appreciate the authors' response. As detailed in the "final justification", I lean toward accepting this paper and have raised my score.

---

> > > ### Author Response · Authors · 2025-08-07
> > > **Thank you**
> > >
> > > Thank you for your positive feedback! We are glad that our response addressed your concerns. We will add the new discussions to the final version.

---

### Note · Authors · 2025-08-13

We sincerely thank all reviewers for their valuable comments and ACs for organizing this nice review process. During the rebuttal period, **we have addressed most of the concerns from reviewers**. Here, we further highlight the contributions of our work and summarize the final revision.

## Significance and contribution
1. Significance: Large diffusion Transformers are compute-intensive and lack a principled and scalable HP selection method, which blocks the progress on vision foundation models. Our work gives a preliminary solution to this significant problem.
2. Contribution: We give a rigorous µP form for diffusion Transformers (beyond heuristics) and validate the effectiveness of µP across architectures and data scales (**up to 18B parameters**), which provides the vision community a principled approach for scaling diffusion Transformers efficiently.

## Final revision

### New results
1. We add the results of the learning rate transfer of PixArt-α-µP, which confirms that the base learning rate searched on the 0.04B proxy model can be transferred to a larger model.
2. We add the GenEval evaluation result of MMDiTs, which further supports the claim that MMDiT-µP achieves better performance at inference.

### Writting
1. We make the experiments for verifying the transferability of other HPs clearer.
2. We clarify that we unify the µP of diffusion Transformers and the classic µP by leveraging the NE⊗OR⊤ Program framework.
3. We improve the visualization to highlight the convergence speed improvements enabled by µP.
4. We emphasize that our current results already validate that µP works well without CFG (DiT) and with CFG (PixArt-α, MMDiT).
5. We revise the text to define REPA weight and warm-up iteration more explicitly.
6. We describe the selection of the optimal hyperparameters in MMDiT experiments more clearly.

### Discussion
1. We add more discussion on the significance and contribution of our work.
2. We cite the blog mentioned by Reviewer AKwD.
3. We add a discussion on the extension of µP to other learning schedules and different depths.
4. We discuss how our assumptions can be relaxed to reflect the practical models better.
5. We add more discussion on the investigation for the optimal design of proxy tasks.
6. We add more discussion on why we chose to scale the model width by increasing the number of attention heads.
7. We add more discussion on the relation between the optimal learning rate and the stability edge.

---

### Decision · Program_Chairs · 2025-09-17

**Decision:**

Accept (poster)

**Comment:**

This paper extends Maximal Update Parametrization (µP) to diffusion Transformers with both theoretical grounding and large-scale empirical validation. It demonstrates robust hyperparameter transfer across architectures (DiT, PixArt-α, MMDiT), reducing tuning cost by over 90% with competitive performance. While some reviewers noted limited novelty, others praised its rigor, broad applicability, and practical impact. The rebuttal addressed concerns thoroughly with new experiments, clearer discussions, and stronger justification. Overall, the paper offers a grounded approach to scaling diffusion models efficiently. I recommend acceptance.